# Countering reproducibility issues in mathematical models with software engineering techniques: A case study using a one-dimensional mathematical model of the atrioventricular node

**Christopher Schölzel**[1,4]*, **Valeria Blesius**[1,4], **Gernot Ernst**[2,3], **Alexander Goesmann**[4], **Andreas Dominik**[1]

**1** Technische Hochschule Mittelhessen—THM University of Applied Sciences, Giessen, Germany, **2** Vestre Viken Hospital Trust, Kongsberg, Norway, **3** University of Oslo, Oslo, Norway, **4** Justus Liebig University Giessen, Giessen, Germany

* christopher.schoelzel@mni.thm.de

**Data Availability Statement:** All model files, simulation data, and scripts are available from Zenodo (https://doi.org/10.5281/ZENODO.4775302),

## Abstract

One should assume that in silico experiments in systems biology are less susceptible to reproducibility issues than their wet-lab counterparts, because they are free from natural biological variations and their environment can be fully controlled. However, recent studies show that only half of the published mathematical models of biological systems can be reproduced without substantial effort. In this article we examine the potential causes for failed or cumbersome reproductions in a case study of a one-dimensional mathematical model of the atrioventricular node, which took us four months to reproduce. The model demonstrates that even otherwise rigorous studies can be hard to reproduce due to missing information, errors in equations and parameters, a lack in available data files, non-executable code, missing or incomplete experiment protocols, and missing rationales behind equations. Many of these issues seem similar to problems that have been solved in software engineering using techniques such as unit testing, regression tests, continuous integration, version control, archival services, and a thorough modular design with extensive documentation. Applying these techniques, we reimplement the examined model using the modeling language Modelica. The resulting workflow is independent of the model and can be translated to SBML, CellML, and other languages. It guarantees methods reproducibility by executing automated tests in a virtual machine on a server that is physically separated from the development environment. Additionally, it facilitates results reproducibility, because the model is more understandable and because the complete model code, experiment protocols, and simulation data are published and can be accessed in the exact version that was used in this article. We found the additional design and documentation effort well justified, even just considering the immediate benefits during development such as easier and faster debugging, increased understandability of equations, and a reduced requirement for looking up details from the literature.

BioModels (https://www.ebi.ac.uk/biomodels/
MODEL2102090002), and GitHub (https://github.
com/CSchoel/inamo).

**Funding:** The author(s) received no specific
funding for this work.

**Competing interests:** The authors have declared
that no competing interests exist.

## 1 Introduction

Mathematical modeling in systems biology, along with many other fields, is facing a reproducibility crisis [1, 2]. According to Stodden *et al.* [3], only an estimated 26 Curators of the Bio-Models database recently found that of 455 ordinary differential equation (ODE) models, only 51 As a single extreme case, Topalidou *et al.* [4] reported requiring three months to reproduce a neuroscientific model of the basal ganglia. The situation is similar to the reproducibility issues in wet-lab experiments, but it is less understandable, since *in silico* experiments only involving mathematical models are inherently free of the biological variations that complicate their wet-lab counterparts.

When talking about reproducibility, it is important to clearly define this term [5]. We follow the terminology of Goodman *et al.* [6], with the following modeling-specific adaptations: *Methods reproducibility* is achieved if the same code can be used with the same simulation tools and settings to produce the same results as the original study. *Results reproducibility* is achieved if the model can be rebuilt in a different language, with a different architectural structure, or simulated with different simulation tools using the same experiment protocol to achieve results that closely match those of the original study. *Inferential reproducibility* does not concern the reproduction of simulation data, but the reproduction of the conclusions drawn from the analysis of that data and the properties of the model. For the most part of this article we will not talk about inferential reproducibility, as our focus lies on model design and not on biological findings.

A lack in methods and results reproducibility can have direct consequences for the usefulness of a model. One example of this is the one-dimensional mathematical model of the atrioventricular (AV) node by Inada *et al.* [7]. It has been labeled as "ground-breaking" [8], because it was the first detailed model of the AV node, and it is still the only AV node model among over 600 models in the Physiome Model Repository [9]. We chose it for our research, because it is able to simulate many important phenomena of the cardiac conduction system including AV nodal reentry while still remaining manageable in its own complexity. Moreover, the article comes with a supplement that contains not only simulation results of the full model but also a set of figures that show the characteristics of the individual ion channel and ion pump models with up to eight individual plots for a single channel. Despite these indicators for a high quality article, the methods of Inada *et al.* are unfortunately not reproducible as there is no executable code available that can produce the results of the original study and reproducing the results with a reimplementation in another language took us more than four months. It seems intuitive that such difficulties in reproducibility may lead to fewer reproduction attempts and therefore less scientific impact. The issues we encountered with the Inada model may have prevented its widespread application, and thus, to some extent, hindered scientific progress in cardiovascular modeling. Given the number of such cases, we believe that it is unlikely that they arise out of a lack of scientific rigor. In contrast, it seems that the inherent complexity of such models inevitably opens the door to human error and that new tools and workflows are required to manage this complexity.

Researchers have already proposed several approaches to increase reproducibility in mathematical modeling. The most pressing and obvious suggestion is to publish the full simulation code, including executable scripts that produce the simulation results and plots that appear in the corresponding article [1–3, 10–15]. Many also advocate the use of literate programming in the form of electronic notebooks that mix textual descriptions and code as publication format [1, 4, 10–13, 16]. However, Medley *et al.* [16] also note that electronic notebooks can be too rigid for the creation of large and complex models and pose some difficulties for version control. Along with the code, data used for plots in the article should also be published, including

the simulation output of the published model [1–3, 10, 14]. Both data and code could be stored in specialized model databases that allow model discovery via semantic information [1, 2, 12]. Workflow systems such as Galaxy [17] or KNIME [18] can be used to publish simulation procedures in a format that ensures methods reproducibility through the use of standardized components [1, 10, 16].

Other suggestions concern the role that academic journals can play in ensuring and promoting reproducibility. Publication checklists [1, 11] or a "seal of approval" [1, 11, 14, 15] could provide missing incentives for researchers to put more focus on all forms of reproducibility. A few journals even already experiment with publication workflows that aim to guarantee methods reproducibility of published models. *PLOS Computational Biology* recently started a promising pilot project in collaboration with the Center for Reproducible Biomedical Modeling [19], which extends the peer review with an additional step in which reviewers specifically evaluate the methods reproducibility of the computational modeling aspects of a submission [15]. The journal *Physiome* takes a similar approach by publishing articles that demonstrate the consistency and reproducibility of mathematical models already described in other publications. Here, too, the actual methods reproducibility is assessed by independent *Physiome* curators.

Apart from these suggestions, which are specific for mathematical modeling and/or systems biology, researchers also advocate for the application of common best practices from software engineering. This includes structured documentation [1, 3, 11, 12], version control [10–13, 16], unit testing [2, 12, 13, 16], the use of open standards [1, 2, 12, 14], human-readable code with style guides [2, 13], modularity [11, 13], object-orientation [12, 13], the use of virtual machine specifications [1, 10, 11], and the long-term archival of code [2].

Borrowing software engineering concepts for improving the methods and results reproducibility of mathematical models seems natural, since these models are, after all, software. As models grow in size and complexity towards examples such as the *Mycoplasma genitalium* whole-cell model by Karr *et al.* [20] or the central metabolism of *E. coli* by Millard *et al.* [21], they face the same kind of issues that software faces when it evolves from a single script of a few lines of code to a complex system with thousands or millions of lines of code. While these issues started to appear only fairly recently in systems biology, they are known for decades in software engineering and efficient solutions have been and are still being developed. Hellerstein *et al.* [13], therefore, argue that modelers should rethink their work as "model engineering" by applying software engineering techniques to the domain of mathematical modeling.

In our attempt to make the Inada model more reproducible, we build on the ideas of model engineering and our own previous work. Most importantly, we found that languages that are modular, descriptive, human-readable, open, graphical, and hybrid (MoDROGH) can help to increase both methods and results reproducibilty as well as reusability, extensibility and understandability [22]. We verified the effectiveness of the consistent use of these characteristics by creating and analyzing a modular version of the Hodgkin-Huxley (HH) model of the squid giant axon [23]. Since the Inada model mainly consists of HH-type ion channels, it is highly likely that this model can also benefit from our design approach. While implementing the HH model, we also developed a workflow with unit tests that are run automatically on an online server every time the code is updated. This concept is called continuous integration (CI) in software engineering and was developed precisely to ensure that software can be installed and run in an environment that is completely separate from the development environment [24]. It is already used, for example, in the bioinformatics framework NF-CORE [25], and in the OpenWorm project, which aims to model *Caenorhabditis elegans* [26]. We expect that this, combined with a model architecture that follows the MoDROGH guidelines, and regression

tests, which ensure that changes to a model do not affect the simulation output, can solve many if not all the reproducibility issues present in the Inada model.

We believe that results from this case study will be applicable to a large set of systems biology models for several reasons: The Inada model is a good example of the range of difficulties and pitfalls one faces when trying to ensure the reproducibility of methods and results of an *in silico* study. Inada *et al.* certainly tried to make their work as transparent as possible. Yet still, the model exhibits all the common reproducibility issues identified by the BioModels reproducibility study [27]—"recoverable" issues like sign errors, missing equations, order of magnitude, and unit errors, as well as "non-recoverable" issues such as missing parameter values, missing initial values and errors in equations. It is also a representative example for challenges in reproducing models in a multi-scale context. On the one hand, the full one-dimensional model of the AV node is in itself a multi-scale model, since it covers cell and organ scales with observed effects ranging from milliseconds to seconds. On the other hand, all 6 published reproductions of the results of Inada *et al.* include the single AV cell model in a larger multi-scale context, be it a 3D heart model [28–30], the cardiac conduction system [31], or a one-dimensional ring model of the sinoatrial (SA) node [32, 33]. Finally, none of the techniques and guidelines that we apply are specific to the Inada model or electrophysiological models in general. The MoDROGH criteria were already applied to an organ-level model of the human baroreflex [22] and both CI and regression tests are concepts borrowed from software engineering, which are applicable to any piece of software. This should also allow to transfer our results to other MoDROGH languages like the Systems Biology Markup Language (SBML) [34] or CellML [35].

We therefore address the following research questions:

**RQ1** What are the factors that hinder the reproduction of the methods and results of the Inada model?

**RQ2** Are software engineering techniques (in particular a MoDROGH design, regression tests, and CI) suited to overcome the issues identified in RQ1?

We will answer these questions by first giving an overview of the Inada model, the resources available for reproduction, and our design philosophy for the reimplementation in Section 2. We then describe all reproducibility issues along with our solutions in Section 3. In Section 4 we discuss the answers to our research questions as well as the general applicability of the techniques that we presented and their limitations. Finally, we draw our conclusion in Section 5.

## 2 Materials and methods

### 2.1 The Inada model

The one-dimensional mathematical model of the atrioventricular node (AV node) by Inada *et al.* [7], which we simply call the Inada model in the following, consists of a one-dimensional chain of different cell types: For the sinoatrial node cells and the atrial cells, preexisting models are used, but for the atrionodal (AN), nodal (N), and nodal-His (NH) cells, the authors developed own formulations. In total these three new cell types are composed of eight ion channels, two ionic pumps, and four compartments with variable $Ca^{2+}$ concentrations:

- ion channels

  - background channel ($I_b$)

  - L-type calcium channel ($I_{Ca,L}$)

- rapid delayed rectifier channel ($I_{K,r}$)

- inward rectifier channel ($I_{K,1}$)

- sodium channel ($I_{Na}$)

- transient outward channel ($I_{to}$)

- hyperpolarization-activated channel ($I_f$)

- sustained outward channel ($I_{st}$)

- ion pumps

  - sodium calcium exchanger ($I_{NaCa}$)

  - sodium potassium pump ($I_p$)

- compartments containing variable $Ca^{2+}$ concentrations

  - cytoplasm ($[Ca^{2+}]_i$)

  - junctional sarcoplasmic reticulum (JSR) ($[Ca^{2+}]_{jsr}$)

  - network sarcoplasmic reticulum (NSR) ($[Ca^{2+}]_{nsr}$)

  - "fuzzy" subspace ($[Ca^{2+}]_{sub}$), which is the "functionally restricted intracellular space accessible to the $Na^+$/$Ca^{2+}$ exchanger as well as to the L-type $Ca^{2+}$ channel and the $Ca^{2+}$-gated $Ca^{2+}$ channel in the SR" [7, 36]

- concentrations assumed to be constant

  - extracellular calcium concentration ($[Ca^{2+}]_o$)

  - intra- and extracellular sodium concentrations ($[Na^+]_i$, $[Na^+]_o$)

  - intra- and extracellular potassium concentrations ($[K^+]_i$, $[K^+]_o$)

## 2.2 Available material

In the first stages of our reimplementation of the Inada model, we relied only on publicly available data. This included the article by Inada *et al.*, the supplementary data for this article in PDF format, and the CellML version of the model, which was created by Lloyd [37] and published in the Physiome Model Repository [9]. The CellML implementation contained code that was not in the paper and did not produce simulation output that resembled any of the plots in the original article. We therefore used it as a reference, but did not rely on its correctness. We supervised two Bachelor's theses that reimplemented the CellML model in Octave and in Modelica. Both projects were able to reproduce some but not all the reference plots in [7]. Before we implemented the current version, we therefore attempted to obtain the original C++ implementation of Inada *et al.* by contacting the authors themselves and the editors of the Biophysical Journal. Unfortunately, we did not receive an answer and our attempt to contact Lloyd for comments on the CellML model was equally unsuccessful. In a second attempt at a later stage of the development, we reached out to the production team of the Biophysical Journal and to the general help address of the Physiome Model Repository. The former finally allowed us to obtain the C++ code and the latter clarified some questions about the CellML implementation and improved our confidence in this code.

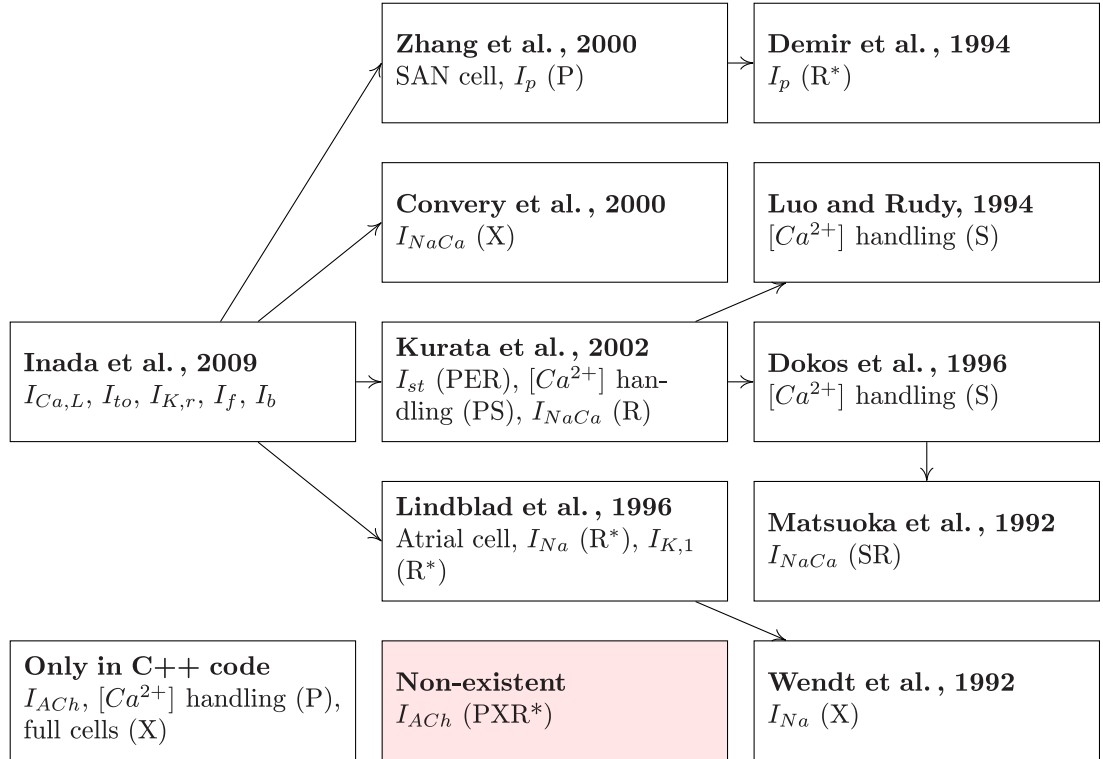

**Fig 1. Tree of references that we traversed to obtain all relevant information to understand and test the model.** Arrows between nodes indicate that the article at the beginning of the arrow cites the article at the end of the arrow. Each node contains a list of model parts that could only be reproduced by using this reference. This is further specified by distinguishing if the reference was needed to determine parameter values (P), correct errors (E), untangle equation semantics for modularization (S), obtain an additional (R) or the only available (R*) reference plot, or to reproduce the experiment protocol (X).

## 2.3 Implementation process

Due to the discrepancies between the article, the CellML code, and the C++ code, we decided to implement the components of the Inada model one by one, testing each component before moving on to the next. In order to obtain reference plots, experiment protocols and parameter values as well as to understand the equations deeply enough to bring them into a modular structure, we needed to examine a total of nine additional articles that were cited directly or indirectly in the Inada model. The full tree of references can be seen in Fig 1. Additionally, an estimation of the time spent on research, implementation, testing, bug fixing, and refactoring and documentation can be seen in Fig 1 in S1 Text.

## 2.4 Model design

Our design philosophy was based on our own guidelines established for using the MoDROGH criteria of suitable modeling languages for systems biology, which can improve the methods and results reproducibility, understandability, reusability, and extensibility of models [22, 23]. In short, this includes the following design goals: The model should follow a modular design with small self-contained modules with clearly defined, minimal interfaces. Each module should only represent a single physiological compartment or effect. The code should be DRY (for "don't repeat yourself"), meaning that parts of the code that have similar structure are only implemented once and then reused at the respective position. Equations structure and variable names should convey their meaning, and should not be adjusted for brevity or

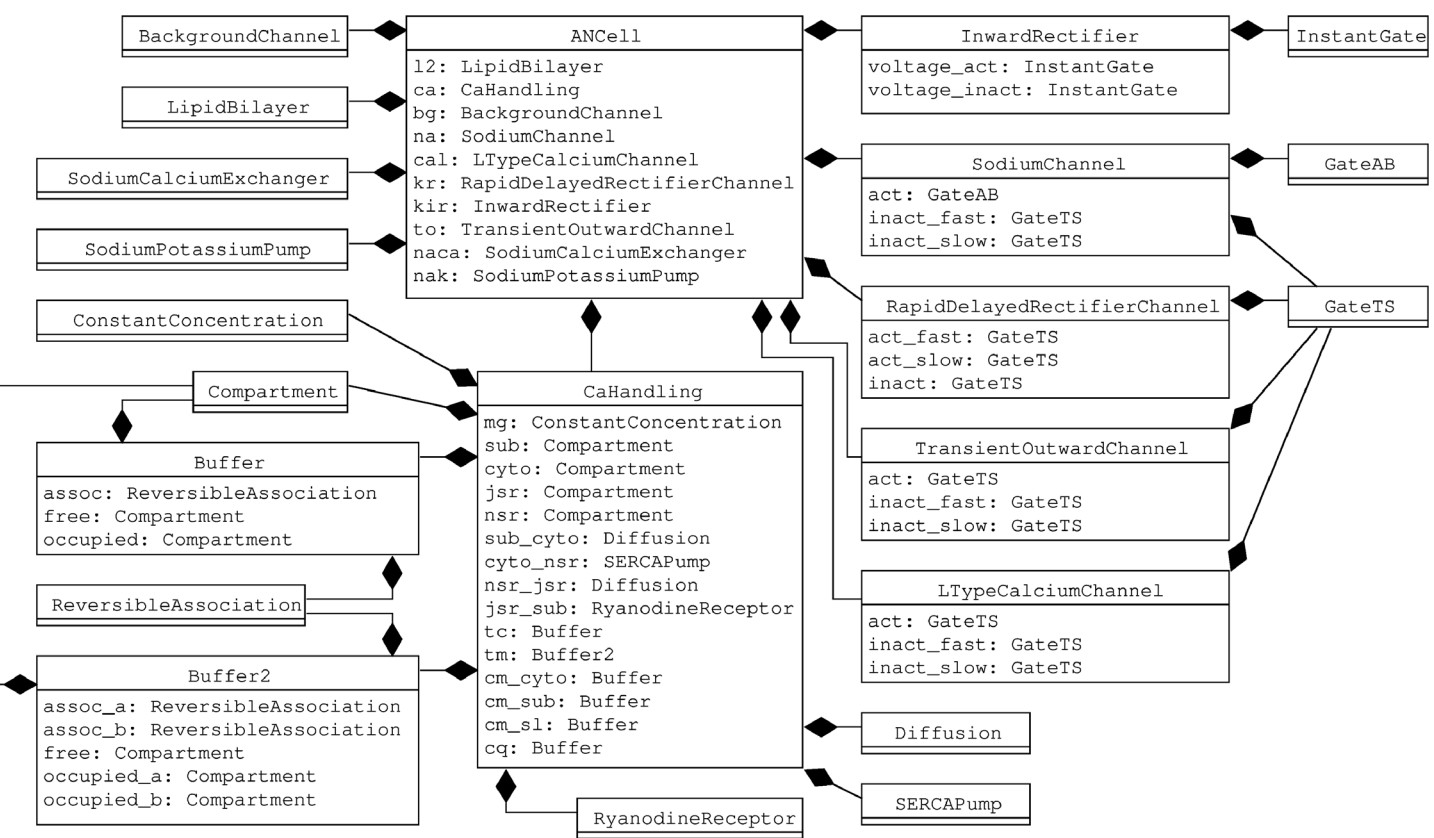

**Fig 2. Hierarchical composition of AN cell model in InaMo version 1.4.1 as UML diagram.** The composition arrow ——◆ indicates that the model at the beginning of the line is a part of the model at the end of the line, where the diamond shape is located.

perceived ease of implementation. All variables and parameters should have International System of Units (SI) units and should be documented with at least a short sentence that explains what they represent. The model should also have a graphical representation that explains the modeled system using symbols that reflect the biological appearance or function of components. This representation should be adjusted manually for understandability, and it should be tied to the model equations to guarantee correctness and completeness. Additionally, the model code should be available in an online repository. In our case the repositories are GitHub (https://github.com/CSchoel/inamo), Zenodo [38], and BioModels (https://www.ebi.ac.uk/biomodels/MODEL2102090002).

We use the same basic model structure as in a previous work, where we tested the viability and benefits of building an electrophysiological model with these guidelines by implementing the Hodgkin-Huxley model [23]. The cell models consist of a model of the lipid bilayer, a number of ion channels with a common base class, and separate models for voltage or current clamp experiment protocols. The ion channel models again contain smaller modules that represent individual gating variables. All parts of the model are connected with a basic interface for components in an electrical circuit diagram with the convention that the positive pin resides on the outside of the cell while the negative pin is on the inside. An example of the full composition structure of the AN cell model can be found in Fig 2. Like in the HH model, we also used the modeling language Modelica [39], since it implements the MoDROGH criteria to a large extent and lends itself well to the application of software engineering techniques

because it is an established industry standard. However, similar results could also be achieved using, for example, SBML [34], CellML [35], or Antimony [40].

## 2.5 Software versions

For our implementation we used OpenModelica version 1.16.0 [41] as Modelica compiler and integrated development environment (IDE) and also version 0.7.2 of Mo—E [42] with the corresponding plugin for Atom version 1.49.0. For our test scripts we used Julia version 1.4.2 [43] with version 1.1.0-alpha.3 of our own library ModelicaScriptingTools.jl [44]. We keep the code under version control using Git [45] version 2.28.0 in a repository hosted on GitHub [46]. We also use GitHub Actions [47] to run our CI scripts. The plots were produced using Python 3.8.3 [48] with the plotting library matplotlib version 3.1.2 [49, 50]. Icons were created using Inkscape version 1.0 [51] with our extension MoNK version 0.2.0 that converts the Inkscape vector graphics to Modelica annotation format [52]. The CellML model was analyzed using OpenCOR version 0.6 [53].

## 3 Results

This section is structured according to the issues that hindered our reproduction of the results of the Inada model. For each issue we first explain the problem in detail and then show how it is solved in our Modelica reimplementation, which we will call InaMo in the following.

## 3.1 Missing equations and parameters

**3.1.1 Problem description.** The first and most obvious issue with results reproducibility of the Inada Model are parameter values and equations that are missing in the article, which are listed in Table 1. An example is the acetylcholine-sensitive potassium channel. The whole channel equations, as well as the influence of acetylcholine on $I_f$ and $I_{Ca,L}$, only exist in the

**Table 1. Missing information in the Inada model including all parameters, equations and starting values that cannot be found in the original article.**

| Component | Affected part | Recoverable from |
|---|---|---|
| $I_{ACh}$ | all equations | C++ code |
| $I_{ACh}$ | parameter $[ACh]_i$ | not recoverable |
| $I_f$ | ACh-sensitive term in equation | C++ code |
| $I_{Ca,L}$ | ACh-sensitive term in equation | C++ code |
| $I_{Ca,L}$ | parameter ach_l | not recoverable |
| $[Ca^{2+}]$ handling | equation for $V_{cell}$ w.r.t. $C_m$ | C++ code |
| $[Ca^{2+}]$ handling | parameter $SL_{tot}$ | C++ code |
| $[Ca^{2+}]$ handling | all parameters but $SL_{tot}$ and $V_{cell}$ | [54] |
| $I_{st}$ | parameter $E_{st}$ | [54] |
| $I_{K,r}/I_{K,1}/I_{to}$ | parameter $E_K$ | calculated from $[K^+]_i$, $[K^+]_o$ |
| $I_p$ | parameters $K_{m,Na}$, $K_{m,K}$ | [55] |
| $I_p$ | parameter $\bar{I}_p$ | called $I_{p,max}$ in [7, S15] |
| $I_{Na}$ | parameter $P_{Na}$ | called $g_{Na}$ in [7, S15] |
| $I_{to}$ | starting values $r$, $q_{fast}$, $q_{slow}$ | called $q$, $r_{fast}$, $r_{slow}$ in [7, S16] |
| $I_{NaCa}$ | parameters $K_x$ [1] and $Q_y$ [2] | [56, 57] |

[1] $x = ci, cni, 1ni, 2ni, 3ni, co, 1no, 2no, 3no$

[2] $y = ci, co, n$

C++ code. Neither the equations nor the parameters are mentioned in the article, and we are not aware of any description in subsequent articles of the authors. The C++ code also does not give a value for $[ACh]_i$, which is set to zero in the CellML version. It is possible that this was a planned extension, which was never realized and not used for the plots in the article, but with the available material it is impossible to tell whether that is the case. Other parameters are missing in the article, but could be recovered from cited literature or the C++ code. There were also parameters that were hard to find due to naming confusions. One example that caused severe errors for us was that the value given for $g_{Na}$, the conductivity of $I_{Na}$, is actually the value for the permeability $P_{Na}$ that is used to calculate $g_{Na}$. As a last minor piece of missing information, the article does not specify how the avoidable discontinuities in equations 1 and 14 of $I_{Na}$ in table S3 should be handled.

**3.1.2 Solution 1: Continuous integration.** To ensure that such omissions do not hinder reproduction of simulation results, it is not enough to rely on human diligence. With a total of 85 parameter values, there is a statistical argument to be made about the expected percentage of errors that a single author or reviewer might be able to spot. Regardless of how the actual numbers would turn out, it does not seem reasonable to expect or demand 100.

Such a guarantee is only possible, if the complete code that is required to run the simulation on a different machine is published alongside with the model. Inada *et al.* did publish parts of their code, but not the full version, which left us with some open questions regarding the acetylcholine-sensitive potassium channel. In contrast, the CellML model is complete, but based on errors that we found in the code one must assume that simulations were only performed with the N cell model and not with the other two cell types.

For the new implementation, we therefore not only publish the full model definition but also the scripts that we used for simulation and plotting. To ensure that the published code is complete and does also work on other machines, we used the CI service GitHub Actions [47], which is free for public open source projects. For each update of the code, a build in a fresh virtual machine is started on the GitHub servers, which downloads the new release and runs the simulation script. The current build status can be indicated to users with a small badge in the repository, and if a build fails, the programmer is informed via e-mail. This mechanism guarantees that the repository contains everything that is required to perform simulations on a machine that is physically separated from the original development environment, i.e. it guarantees methods reproducibility. The build scripts for CI services such as GitHub Actions are easy to write and provide the additional benefit that they have to contain a full description of the development environment including installation scripts and non-standard software dependencies. The build script that we used for our implementation of the Inada model can be found in Listing 1.

**Listing 1**. CI script for InaMo version 1.4.3 using GitHub Actions. The script creates a virtual machine running the Ubuntu operating system, installs Julia, OpenModelica, the Modelica Standard Library, and required Julia packages, and runs the unit tests defined in the file scripts/unittests.jl. It runs automatically whenever a new commit is pushed to the main branch of the Git repository.

```
on:
  push:
    branches: [ main ]
    tags: 'v*'
  pull_request:
    branches: [ main ]
jobs:
  build:
    runs-on: ubuntu-latest
```

```
steps:
- uses: actions/checkout@v2
  with:
    submodules: true
- uses: julia-actions/setup-julia@v1
  with:
    version: 1.6
- uses: THM-MoTE/setup-openmodelica@v1
  with:
    version: 1.17.0
- name: Install Modelica standard library
  run: sudo apt-get install omlib-modelica-3.2.3
- name: Install Julia requirements
  run: |
    export PYTHON="''
    julia --project=. -e 'using Pkg; Pkg.instantiate()'
- name: Run unit tests
  run: julia --project=. scripts/unittests.jl
```

**3.1.3 Solution 2: Version control.** Even if the complete code of a model is published, an exact reproduction of methods might still fail, because of changes that have been added to the code after the model was published. It might even be the case that a figure in an article was created with a newer or older version of the code than other figures. One such uncertainty about code versions is the question if the current $I_{ACh}$ was included and activated in the simulations performed by Inada *et al*. The C++ code gives some clues as it contains a list of major changes with the date of the change. According to this information, $I_{ACh}$ was added on 11/04/2008, which is before the initial submission to Biophysical Journal on 27/02/2009. However, this is still not enough to be sure that $I_{ACh}$ was used for simulations, because another change—a rescue effect for $I_{Ca,L}$—was added on 23/10/2008, but the current parameter values used in the published version clearly disable it. If the code was under version control and the history was published, it would be possible to answer this question at least with some confidence by tracking the changes through time.

We therefore publish our reimplementation on GitHub [46], which uses the version control software Git [45]. Additionally, we keep a human-readable log of major changes in a Markdown-formatted [58] text file called CHANGELOG.md in the repository. The simulation results in S2–S32 Figs are tagged with the actual version used for the simulation.

Version control also has several other benefits beyond understanding when, how, and why a model has been changed. Most prominently, it allows researchers to work on a model collaboratively and to merge changes made by different authors, which will become more important in systems biology as models grow in size and models by different groups have to be integrated into a single project. Additionally, version control facilitates debugging by allowing to effortlessly roll back changes to identify the exact edit that introduced an error in the code. Finally, changes that may have to be reverted, like the rescue effect for $I_{Ca,L}$, can be developed on separate branches, which can then either be abandoned or merged into the main branch, depending on whether the feature was deemed beneficial or not.

### 3.2 Errors in equations and parameters

Apart from missing information there are also errors both in equations and parameter values, which can be seen in Table 2. These are typical oversights including sign errors and order of magnitude errors related to unit conversion. An example of a sign error is the erroneous negative sign for $Q_n$ in equation 5 of $I_{NaCa}$ in table S10. Order of magnitude errors can, for example, be found in the parameters $k_x$ and $k_{b_x}$ where $x = f_{TC}, f_{TMC}, \ldots$ in the equations for $[Ca^{2+}]$

**Table 2. Errors in the published equations of the Inada model including wrong signs, shifted floating points, and missing unit conversions.**

| Component | Affected part | Kind of error/correction |
|---|---|---|
| $[Ca^{2+}]$ handling | table S12, eq. 5 | $f_{CMi}$ and $f_{TC}$ must have negative sign |
| $I_{NaCa}$ | table S10, eq. 5 | $Q_n$ must have positive sign |
| $I_{st}$ | table S8, eq. 2 | second occurrence of $V$ must be negative |
| $[Ca^{2+}]$ handling | parameters[1] $k_x$, $k_{b_x}$ | must be multiplied by 1000 ($ms^{-1} \to s^{-1}$) |
| $I_{st}$ | variables $\tau_{qa}$, $\tau_{qi}$ | must be divided by 1000 (ms $\to$ s) |
| $I_{to}$ | variable $\tau_{q_{fast}}$ | constant 0.1266 must be 0.01266 instead |

[1] $x = f_{CM}, f_{CQ}, f_{TC}, f_{TMC}, f_{TMM}$

handling, which all have the unit $ms^{-1}$ in [54] but need to be multiplied by 1000 since [7] uses seconds as unit of time.

As a second type of errors, there are inconsistencies between parameter values in the article and in the C++ and CellML implementations, which can be found in Table 3. Some of them, like the value used for $P_{rel}$, seem to be undocumented changes that do have a large qualitative effect on simulations. Others, like the value of 227700 instead of 22**2**700 for $k_{f_{TMC}}$, seem to be simple typing errors and oversights. Of these, it is notable that in the CellML model, the value of $C_m$ is switched between the NH and the N cell model, which should have stood out if the model was verified in simulations.

These inconsistencies are even more pronounced in the initial values. Due to the large number of initial values to keep track of, we do not show them here but only in Tables 1–3 in S1 Text. It seems as though the initial states were chosen to resemble the near-steady state achieved right before a pulse during a long-term simulation with a current pulse protocol. However, this information is missing in the article. We only found this pattern through trial and error and by a hint in the version history of the CellML model. Even with this knowledge, still large inconsistencies remain between the article, the C++ model, and the CellML model with some open questions. For example, we suspect that there is an order of magnitude error in the reported initial value for the parameter $f_{TC}$ in the N cell model in the article. While these errors do affect simulations for up to 20 seconds, all variables that need an initial value do

**Table 3. Parameter values that differ between the published article and the C++ and CellML implementations.**

| Parameter | Cell type | Unit | Value in article | Value in C++ | Value in CellML |
|---|---|---|---|---|---|
| $P_{rel}$ | AN, NH | $s^{-1}$ | 5000 | 1805.6 | 1805.6 |
| $P_{rel}$ | N | $s^{-1}$ | 5000 | 1500.0 | 1500.0 |
| $V_{cell}$ | AN, NH | $m^3$ | $3.500 \cdot 10^{-15}$ | $4.398 \cdot 10^{-15}$ | $4.400 \cdot 10^{-15}$ |
| $V_{cell}$ | N | $m^3$ | $3.500 \cdot 10^{-15}$ | $3.189 \cdot 10^{-15}$ | $3.190 \cdot 10^{-15}$ |
| $C_m$ | AN | F | $4.0 \cdot 10^{-11}$ | $4.0 \cdot 10^{-11}$ | $4.0 \cdot 10^{-11}$ |
| $C_m$ | N | F | $2.9 \cdot 10^{-11}$ | $2.9 \cdot 10^{-11}$ | $4.0 \cdot 10^{-11}$ |
| $C_m$ | NH | F | $4.0 \cdot 10^{-11}$ | $4.0 \cdot 10^{-11}$ | $2.9 \cdot 10^{-11}$ |
| $k_{f_{TC}}$ | all | $\frac{1}{mM \cdot s}$ | 534 | 534 | 543 |
| $k_{f_{TMC}}$ | all | $\frac{1}{mM \cdot s}$ | 227700 | 222700 | 227700 |
| $E_{st}$ | N | mV | 37.4 | 37.4 | -37.4 |

For $C_m$ the reference article is [7], for all other parameters it is [54].

gravitate towards a steady state, meaning that these issues should not affect results or inferential reproduction.

**3.2.1 Solution 1: Testing.**   Again, it becomes clear that neither authors nor reviewers can be expected to find every single oversight within well over a hundred equations and parameters. Like with missing information, automated tests are the only way to reliably ensure that a piece of code of the size of the Inada model is free of errors. These tests can be run in a CI environment, as described above, which means that each version of a model will be automatically evaluated for errors that may have been introduced accidentally. For InaMo, we use three kinds of automated tests, which are common in software engineering: unit tests, integration tests and regression tests.

Unit tests are fine-grained tests for small software modules. In order to create unit tests, these modules must be independent of the rest of the code. They are designed to pin down errors to a single module and therefore increase the confidence in the correctness of these modules. In mathematical modeling this is important, because often researchers do not want to reproduce the results of the full model but only a part of it. This becomes considerably easier if there is an existing test case for the part that should be reused. InaMo contains unit tests for each individual current and gating variable in the model as well as for the diffusion reactions, ryanodine receptor, SERCA pump, and buffer components in the $[Ca^{2+}]$ handling. Each of the experiment models used for this test import the exact same components used for the full cell model in an isolated environment where all external variables that influence the behavior of the component are carefully controlled. Almost all of these experiments correspond to plots in the original article or cited references (see Table 4). The only exception are the individual components of the $[Ca^{2+}]$ handling, because neither Inada *et al.* [7] nor Kurata *et al.* [54] provide plots at this level of detail.

Moving to the next category, integration tests help to ensure that there is no error in the connection between modules when they are combined to a larger system. They work much in the same way as unit tests, but are applied to the whole software instead of only individual modules. This helps to spot errors that only emerge from the interaction between the individual modules. In InaMo, there are integration tests for each cell type (AN, N, and NH cells) as well as both for constant and for varying $[Ca^{2+}]_i$. This prevents issues like in the CellML model, where probably only N cells were tested.

The third category of tests are regression tests, which ensure that the output produced by a piece of code does not change accidentally. They are typically used, when the output is large and has a complex structure that is otherwise hard to incorporate in unit tests. In mathematical modeling, these kinds of tests can serve three purposes. First, they ensure that changes to a part of a model do not have unforeseen consequences in other parts of the model. Second, they highlight these changes during the development process, which increases the probability that plots and formulas in the corresponding article will be changed accordingly. Third, the mere fact that regression tests require keeping reference output data in the repository helps to preserve results reproducibility in the future, when the software that produces the simulation results may not be available anymore. The repository for InaMo contains a separate sub-repository with reference data for all models used in unit and integration tests and the GitHub Actions script performs regression tests for them. If changes are found, the plotting script can be configured to output additional plots from the reference data so that these changes can be inspected visually. Since Modelica is a human-readable language and the reference data is stored in the human-readable CSV format, researchers in the far future will only need a simple text editor to reproduce the results in this article.

**3.2.2 Solution 2: Unit consistency checks.**   As another category of tests, which is specific to mathematical models, automated unit consistency checks can help to avoid order of

**Table 4. Summary of individual experiments that were reproduced with InaMo.**

| Part | Figure | Exact | Issues/Required changes |
|---|---|---|---|
| $I_{Ca,L}$ | [7, S1] | (✓) | $T_{hold}$ = 5s, $V_{hold} \leftarrow$ -70 mV, NH parameters, S1H: reference timescale must be multiplied by 0.75 |
| $I_{to}$ | [7, S2] | ✘ | $T_{hold}$ = 20s, NH parameters, S2E: current too high, S2D: higher minimum in reference |
| $I_{K,r}$ | [7, S3] | ✘ | $T_{hold}$ = 5s, S3C: reference shifted towards higher voltages |
| $I_f$ | [7, S4] | ✘ | $T_{hold}$ = 20s, S4C: qualitative differences |
| $I_{st}$ | [7, S5] | ✘ | $T_{hold}$ = 15s, $g_{st} \leftarrow$ 0.27 nS, S5C: qualitative differences |
| $I_{st}$ | [54, 4] | ✓ | $T_{hold}$ = 15s |
| $I_{NaCa}$ | [7, S6] | (✓) | $[Ca^{2+}]_o \leftarrow$ 2.5 mM, $[Ca^{2+}]_{sub}$ = 0.00015 mM |
| $I_{NaCa}$ | [54, 17] | ✓ | $k_{NaCa}$ given as $\frac{pA}{pF} \Rightarrow$ multiply by $C_m$ |
| $I_{NaCa}$ | [57, 19] | (✓) | 0.25 nA < $k_{NaCa}$ < 1 nA chosen to fit plots |
| $I_{Na}$ | [65, 2] | ✓ | $T_{hold}$ = 2 s, $T_{pulse}$ = 50 ms, $na_p \leftarrow 2.1 \frac{pl}{s}$ |
| $I_b$ | - | | trivial, no need for test |
| $I_{ACh}$ | - | | no description or plot available |
| $I_{K,1}$ | [65, 2] | ✓ | |
| $I_p$ | [64, 12] | ✘ | reference mixes $I_p$ with background currents |
| $[Ca^{2+}]_i$ | [7, S7] | ✘ | only as part of full cell simulation (see row below) |
| Cell | [7, S7] | ✘ | $T_{hold}$ = 300 ms, $T_{pulse}$ = 1 ms, $I_{pulse}$ = −1.2 nA (AN), $I_{pulse}$ = −0.95 nA (AN), AN, NH: resting potential too high and action potential slightly too short, N: $[Ca^{2+}]_i$ too high |

The table shows the part of the model that is tested with the experiment, a reference to the original figure, the information whether the plot could be reproduced exactly, and a list of issues and changes that were required to obtain a good agreement with the original plot. For a visual comparison of plots, see Figs 2–33 in S1 Text. For the exactness, ✓ means a near perfect reproduction was possible with minimal adjustments, (✓) means that significant changes or manual parameter tuning were required, and ✘ means that even after adjustments only qualitative agreement was achieved while some visible differences remain. In the last column, an equals sign (=) means that a parameter value was not given in the original article and had to be determined by us, whereas an arrow (←) means that the parameter value was given, but had to be changed. The entry "NH parameters" means that we had to use the parameters given for the NH cell model, while Inada *et al.* report that they used parameter settings of the AN cell model.

magnitude errors. In declarative languages like Modelica, SBML, or CellML, variables can be annotated with unit definitions and software tools can track the conversion between units in an equation with symbolic mathematics. A unit consistency check then produces an error or a warning if an equation is found, where the right-hand side has a different unit than the left-hand side. In InaMo, variables have unit definitions according to the SI wherever possible and the tests in the CI script contain consistency checks, which are performed when loading the individual models.

**3.2.3 Solution 3: Modular model structure using object orientation.** Unit tests require model components to be defined in an independent modular structure. It must be possible to run a simulation using only a single component and a minimal experiment setup surrounding it. At the same time, the code of that component must be exactly the same code in the same file that is used in the full cell model, because otherwise the unit test cannot make assertions about the correctness of the full model.

With InaMo, we therefore consequently follow a modular design structure with minimal interfaces between components. Each component is defined in its own file, which is imported both in the unit test of that component and in the full cell model. The full hierarchical composition of the AN cell model can be seen in Fig 2. An example that shows how the component `SodiumChannel` is used both in the unit test `SodiumChannelIV` and in the full cell model `ANCell` is presented in Listing 2.

**Listing 2**. Example for construction of unit tests and full cell model out of the same components. The component `SodiumChannel` is used both in `SodiumChannelIV`, which defines a voltage clamp experiment to test the current-voltage relationship of $I_{Na}$ that is used as a unit test, and in `ANCellBase`, which is the base for the full AN cell model. In both cases, an instance of `SodiumChannel` with the name `na` is defined and then connected to other components in the model using `connect()` equations. Additionally, in `ANCellBase`, the initial values of gating variables are adjusted. The ellipses (...) denote code that is not shown including inheritance from base classes, additional components and connections, and graphical annotations.

```
model SodiumChannelIV "IV relationship of I_Na (Lindblad 1996,
Fig. 2b)"
  ...
  InaMo.Currents.Atrioventricular.SodiumChannel na annotation(...);
  ...
equation
  connect(vc.p, na.p) annotation(...);
  connect(vc.n, na.n) annotation(...);
  ...
end SodiumChannelIV;
model ANCellBase "base model for AN cell type"
  ...
  InaMo.Currents.Atrioventricular.SodiumChannel na(
    act.n.start = 0.01227,
    inact_slow.n.start = 0.6162,
    inact_fast.n.start = 0.7170
  ) annotation(...);
  ...
equation
  ...
  connect(na.p, p) annotation(...);
  connect(na.n, n) annotation(...);
  ...
end ANCellBase;
```

A modular, object-oriented design is not only beneficial because it allows defining unit tests, but it also in itself can help to reduce possible sources of errors by reducing redundancy in the code. For example, the CellML implementation of the Inada model is split into three separate files for the AN, N and NH cells. This means that every error that is found in the model has to be corrected in all three files, leaving the opportunity open for additional oversights. Conversely, the C++ implementation handles all model types in a single file, but this also creates a problem. The code that sets parameter values uses conditional branches based on which cell type should be simulated. Because of the monolithic structure, values need to be defined for each current, even for those currents which are not present in the selected cell type. This led to an error that the parameter $E_{st}$ for the current $I_{st}$ had a wrong sign in the AN and NH cell setup. For the C++ implementation, this is no issue, since $I_{st}$ is only used in the N cell model, where the sign was corrected. However, it appears that this error was accidentally transmitted to the CellML model, where all three cell types have the wrong sign for $E_{st}$.

In InaMo, we follow the DRY (don't repeat yourself) principle of software engineering: Each component and parameter is defined exactly once in the code, reusing common structures as much as possible to reduce redundancies. In an object-oriented language like Modelica, this can be achieved in two ways: First, components can be instantiated, which means that their code is imported into a model under a chosen name. Two instances of a component can have different parameter settings, allowing, for example, to use the same component `GateTS` both for the activation and inactivation gate of an ion channel as shown in Listing 3. Second, models can also inherit components, parameters, and equations from common base classes. This is similar to composition via instantiation, but has the added benefit that the inherited parts directly become a part of the model without the need to access them through a component name. For example, this is very useful for the ion channels in the Inada model, which almost all follow an electric analog. The base class `IonChannelElectric` defines the basic behavior of an ion channel as voltage-dependent resistor with an attached voltage source and can be reused for $I_b$, $I_{ACh}$, $I_f$, $I_{K,1}$, $I_{Ca,L}$, $I_{K,r}$, $I_{st}$, and $I_{to}$. This is shown in Listing 4.

**3.2.4 Solution 4: Specialized testing library for Modelica models.** There are currently not many solutions for automated tests that are specifically designed for mathematical models. To facilitate the creation of such tests as much as possible, new tools are required. One promising approach is to use libraries that can run simulations from within general purpose programming languages such as Python or Julia and to then use the existing capabilities for automated testing that exist in these languages.

We therefore developed the Julia library ModelicaScriptingTools.jl (MoST.jl) [44], which uses the library OMJulia.jl [59] developed by the OpenModelica project [41]. With essentially three relevant lines of code, which can be seen in Listing 5 and 7, the library establishes communication with the OpenModelica compiler (OMC), and then loads a given model, runs a simulation with it and performs a regression test. During model loading and simulation, checks for unit consistency as well as for compiler errors and warnings are performed and any issues are reported with human-readable error messages that include the original compiler message if possible. This means that modelers do not need an in-depth knowledge of the Julia language, or any other programming language, to benefit from thorough automated testing. As shown in Listing 1, they can also set up a CI pipeline for their Modelica project with just two calls to the `julia` executable. If required, however, more fine-grained tests and separate simulation scripts can be defined using the application program interfaces (APIs) of MoST.jl and OMJulia.jl for model inspection and simulation and the testing capabilities of Julia.

An experimental feature of MoST.jl also aims to solve the problem of errors occurring in equation and parameter lists in articles by automatically generating a human-readable documentation of a model. This is based on the function `dumpXMLDAE` in the OpenModelica scripting application programmer interface (API), which generates an eXtensible Markup Language (XML) file containing a flat list of all parameters, variables, functions, and equations in a composite model. The equations are not only listed as code but additionally as content Mathematical Markup Language (MathML), which allows to automatically translate them to presentation MathML, which can be, e.g., rendered in a web browser. Since MoST.jl is written in Julia, we can use the highly extensible documentation generator Documenter.jl to generate an HTML documentation of a model by simply inserting an annotated code-block in a Markdown-formatted text file as shown in Listing 6. This can, again, happen in a CI pipeline, ensuring that there is an accurate human-readable documentation for each version of the model. However, automatic generation of such a documentation from a composite model is not trivial, as variables and functions can have multiple aliases, which introduce clutter that has to be reduced. Additionally, variables have to be grouped to keep the list of equations clear and the variable names in the equations short enough for a visually pleasing presentation. The current

implementation state of this feature is enough to give an idea what is possible, but does not yet produce output that can be used as a supplement in an academic journal. An example for InaMo can be seen at https://cschoel.github.io/inamo/v1.4/models/examples/#Tests-for-I_K,1.

**Listing 3**. Example for composition via instantiation in InaMo. The gating model GateTS implements the generic Hodgkin-Huxley equation. The ion channel model `SodiumChannel` uses two instances of GateTS with different names `inact_fast` and `inact_slow` for fast and slow inactivation. Both instances use different fitting functions to replace the generic placeholder function ftau for the time constant of the gating variable. This reduces both the need to redundantly define the Hodgkin-Huxley equations and fitting functions like `genLogistic` and therefore keeps the code DRY.

```
model GateTS
  import InaMo.Functions.Fitting.*;
  ...
  replaceable function ftau = genLogistic;
  replaceable function fsteady = genLogistic;
  Real n(start = fsteady(0), fixed = true) "ratio of molecules in open
conformation";
  outer SI.ElectricPotential v_gate "membrane potential of enclosing
component";
  ...
equation
  der(n) = (fsteady(v_gate)−n) / ftau(v_gate);
annotation(...);
end GateTS;
model SodiumChannel
  ...
  GateAB act(...);
  function inact_steady = pseudoABSteady(...);
  GateTS inact_fast(
    redeclare function fsteady = inact_steady,
    redeclare function ftau = genLogistic(
      y_min = 0.00035, y_max = 0.03+0.00035, x0=-0.040, sx=-1000/6.0)
  );
  GateTS inact_slow(
    redeclare function fsteady = inact_steady,
    redeclare function ftau = genLogistic(
      y_min = 0.00295, y_max = 0.12+0.00295, x0=-0.060, sx=-1000/2.0)
  );
  Real inact_total = 0.635 * inact_fast.n + 0.365 * inact_slow.n;
equation
  open_ratio = act.n^3 * inact_total;
end SodiumChannel;
```

**Listing 4**. Example for ion channel in InaMo. Most ion channels share a base class `IonChannelElectric` that implements the base equations for the electrical analogy to a conductor coupled to a voltage source. Full ion channel models such as `SustainedInwardChannel` then only have to define the `open_ratio` that determines the opening and closing of the channel in dependence of the gating variables.

```
partial model IonChannelElectric "ion channel based on electrical
analog"
  extends Modelica.Electrical.Analog.Interfaces.OnePort;
  parameter SI.ElectricPotential v_eq "equilibrium potential";
  parameter SI.Conductance g_max "maximum conductance";
  SI.Conductance g = open_ratio * g_max "ion conductance";
  Real open_ratio "ratio between 0 (fully closed) and 1 (fully open)";
equation
```

```
  i = open_ratio * g_max * (v - v_eq);
end IonChannelElectric;
model SustainedInwardChannel "I_st"
  extends IonChannelElectric(g_max = 0.1e-9, v_eq=-37.4e-3);
  GateTS act(...);
  GateAB inact(...);
equation
  open_ratio = act.n * inact.n;
end SustainedInwardChannel;
```

**Listing 5**. ModelicaScriptingTools.jl (MoST.jl) script that loads the model file `src/InaMo/Examples/FullCell/AllCells.mo`, performs simulations according to the simulation parameters read from that file (see Listing 7), places the outputs in the folder out, and performs regression tests if it finds reference data in the directory `regRefData`.

```
using ModelicaScriptingTools
using Test
withOMC("out", "src") do omc
  @testset "Example" begin
    testmodel(omc, "InaMo.Examples.FullCell.AllCells";
refDir="regRefData")
  end
end
```

**Listing 6**. Markdown-formatted text file that is used to generate a HTML documentation of InaMo including the HTML string from the model file itself, a list of all equations rendered as MathML, a list of all functions in Modelica syntax, and a table with all variables and parameters.

```
# InaMo
Documentation for InaMo.
```@modelica
InaMo.Examples.FullCell.AllCells
```
```

## 3.3 Availability of data files

**3.3.1 Problem description.**   Inada *et al*. did originally upload their C++ code to the *Biophysical Journal* with the intent to make it available for download, which is to be commended. However, some unknown issue—maybe an update of the publishing platform—seems to have buried this information as there is currently no download link on the journal website. Only after multiple attempts of contacting both the authors and the journal, we were able to obtain the code from the production team of the journal. We asked them to add a download link to the article page so that other researchers would have easier access to the files but received no answer to our request. As mentioned above, information was missing from the article and some errors in equations and parameters were ultimately only recoverable from the C++ code. Without the code we might therefore not have been able to recreate the full cell models at all. Earlier access to the model code could also have reduced the time that was spent fixing bugs in the code.

**Listing 7**. Experiment annotation of the `AllCells` model, which contains full cell tests for all three model types (AN, N, and NH cells). The parameters `StartTime`, `StopTime`, `Tolerance`, and `Interval` are part of the Modelica language specification, the parameter s for the solver selection is a vendor-specific annotation of OpenModelica and the `variableFilter`, which controls which variables occur in the output file, is a vendor-specific annotation of MoST.jl.

```
model AllCells
  FullCellCurrentPulses an(redeclare ANCell cell);
```

```
  FullCellSpon n(redeclare NCell cell);
  FullCellCurrentPulses nh(redeclare NHCell cell);
annotation(
  experiment(StartTime = 0, StopTime = 2.5, Tolerance = 1e-12,
Interval = 1e-4),
  __OpenModelica_simulationFlags(s = "dassl"),
  __MoST_experiment(variableFilter="(an|n|nh)\\.cell\\.(v|ca\\.(sub|
cyto)\\.c\\.c)")
);
end AllCells;
```

**3.3.2 Solution: Services for long-term archival of code and data.** We believe that while the management of supplemental data is the responsibility of scientific journals, researchers should not solely rely on this system. Journals and their archival systems are more focused on text content than on data and—as this case shows—can fail to preserve this relevant information or to make it accessible for future research.

With InaMo, we therefore used multiple fail-safe options. First, we publish our code on GitHub, which has adopted a "pace layers" strategy [60] for archiving code in multiple redundant databases with the extreme of the GitHub Arctic Code Vault that is designed to store code for a thousand years [61]. Second, to make our code citable and more easily accessible for research purposes, we also use Zenodo, which assigns document object identifiers (DOIs) to archived code and data and stores it in the CERN Data Centre [62]. Although it does not extend to the same time spans as the GitHub Archive Program, Zenodo might be the most suitable solution for data uploads such as reference data for regression tests, as those are not fully covered by GitHub's program. With this setup, the availability of our data does not depend on a single academic journal but is in the hands of multiple institutions that specialize in keeping code and data available for future generations of researchers.

## 3.4 Non-executable code

**3.4.1 Problem description.** Even with both the C++ and CellML implementations of the Inada model available, we could not obtain reference simulation results that we could have used for debugging. The C++ code does not include any file with an executable `main()` function, but only function and variable definitions for the equations and variables of the model. The CellML code *is* executable using OpenCOR, but only the N cell model does produce an action potential with the settings given in the model file. As mentioned in Section 3.2, the N cell model has significant errors in the parameter values of $C_m$ and $E_{st}$, which does not increase our confidence that the model is in a state that allows it to be used as a reference.

This already means that the methods of Inada *et al.* are not reproducible. Without executable code, there is no way to obtain simulation results in the same way as the authors did. Additionally, this also limits the results reproducibility of the model as there is no reference implementation or simulation data against which we could compare our reimplementation for testing and debugging purposes. We could use the plots as data source, but this is more error-prone as we will explain in the following subsection.

**3.4.2 Solution: Continuous delivery.** In software engineering, CI pipelines often also include a distribution stage that compiles an artifact which can be distributed to end users if and only if the testing stage did not produce any errors. This process is called continuous delivery (CD), and it can be used in mathematical modeling to ensure that the code that is submitted to a journal or stored in an archive is indeed both complete and correct. GitHub already automatically adds a ZIP archive including the whole repository content to each tagged version of a repository, which can be enough for small projects. In our case, however, we need an

additional step, since the ZIP archive generated by GitHub by default does not include the content of submodules, which we use to store the data files for the regression tests.

The additional work that is required to generate a distribution of a model, is performed by a CI script. In our case, this involves a call to the `zip` tool on the command line and the use of the predefined actions `create-release` and `upload-release-asset` as can be seen in Listing 8.

CD also provides the opportunity to distribute models not only as source code but also in dedicated exchange formats. Since version 1.4.3, InaMo releases contain an export of the main model `AllCells` as Functional Mock-up Unit (FMU). The FMU format is used in the Modelica ecosystem to increase the interoperability of models and to make models available across tool and language barriers. This means that the release version of the `AllCells` model is not only executable in OpenModelica, but also in any of the over 100 tools that support the Functional Mock-up Interface (FMI) [63].

## 3.5 Missing reference plots and experiment protocols

In order to use unit tests, as suggested in Section 3.2, reference data or plots are required that capture the behavior of a single part of the model and therefore provide target results, which can be reproduced. The data supplement of [7] does contain reference plots for $I_{Ca,L}$, $I_{to}$, $I_{K,r}$, $I_f$, $I_{st}$, and $I_{NaCa}$ as well as voltage and $[Ca^{2+}]_i$ curves for the full cell models. However, reference plots for $I_{K,1}$, $I_{Na}$, $I_p$, and $I_{ACh}$ as well as for the $[Ca^{2+}]$ handling are missing. As shown in Fig 1, reference plots for $I_{K,1}$, $I_{Na}$ and $I_p$ could be obtained from the sources that are cited in [7]. This still leaves the $[Ca^{2+}]$ handling and $I_{ACh}$ without reference.

As shown in Table 4, a complete and error-free experiment protocol was only available for $I_{K,1}$. All other experiments required some form of adjustments and in roughly half of the cases no exact agreement with the original plots could be achieved. A common reason for this is that current-voltage relationships of ion channels are usually determined with a test pulse protocol, of which not all parameters were reported in the articles. In this protocol, the voltage is held at a holding potential $V_{hold}$ for a period $T_{hold}$, after which it is immediately set to a pulse potential $V_{pulse}$ for a duration $T_{pulse}$ followed by another holding period and so on. $V_{hold}$ is gradually increased after each pulse and then plotted against the maximum current obtained in the cycle duration. Inada *et al.* give values for $V_{hold}$, $V_{pulse}$ and $T_{pulse}$, but not for $T_{hold}$. This is relevant, because due to the high time constants of slow activation and inactivation gates, some currents only arrive at a steady state after 20 seconds. If $T_{hold}$ is smaller than this time period, the current during a cycle will also be affected by the previous cycle.

**Listing 8**. GitHub Actions script to automatically draft a GitHub release each time a new tag is encountered in the repository. After downloading the source code with the `checkout` action, the version number is saved in the `RELEASE_VERSION` variable, and a ZIP archive is created with the `zip` tool. Then, the `create-release` action is used to create the release draft on the GitHub website and the ZIP archive is attached to this draft with the `upload-release-asset` action. This version uses the whole content of the file `README.md` as body text for the release. The full script for InaMo version 1.4.3 parses only the recent changes from the `CHANGELOG.md` file and also contains additional code to attach an Functional Mock-up Unit (FMU) export of the model `InaMo.Examples.FullCell.AllCells` to the release, which is not shown here.

```
on:
  push:
    tags:
      - 'v*' # Push events to matching v*, i.e. v1.0, v20.15.10
jobs:
```

```
release:
  runs-on: ubuntu-latest
  steps:
    - uses: actions/checkout@v2
      with:
        submodules: true
    - run: echo "RELEASE_VERSION=${GITHUB_REF#refs/*/}"
>> $GITHUB_ENV
    - run: |
        zip -r inamo-${RELEASE_VERSION}.zip . -x \*.git/\* \*.git
    - uses: actions/create-release@v1
      id: create_release
      env:
        GITHUB_TOKEN: ${{ secrets.GITHUB_TOKEN }}
      with:
        tag_name: ${{ github.ref }}
        release_name: Release ${{ github.ref }}
        body_path: README.md
        draft: true
    - uses: actions/upload-release-asset@v1
      env:
        GITHUB_TOKEN: ${{ secrets.GITHUB_TOKEN }}
      with:
        upload_url: ${{ steps.create_release.outputs.upload_url }} #
reference previous output
        asset_path: ./inamo-${{ env.RELEASE_VERSION }}.zip
        asset_name: inamo-${{ env.RELEASE_VERSION }}.zip
        asset_content_type: application/zip
```

In other cases, reported parameters had to be adjusted manually to obtain a good agreement with the original plots. This includes simple oversights like wrong units, but also cases where it seems that different values were used than those that were reported, as for S1 Fig in [7], where $V_{hold}$ is given as −40 mV, but we achieve much better results with a value of −70 mV. It also seems that Inada *et al.* used the parameter settings of the NH cell model for plots of $I_{Ca,L}$ and $I_{to}$, even though the article states that parameters of the AN cell model were used. Another example are the plots for $I_{NaCa}$ by Matsuoka *et al.*, where it is stated that a scaling parameter was used for each of the individual plots, but the value of the scaling parameter is not given in the article.

As a final issue, we could not obtain isolated reference plots for some of the components as they were only used and reported in combination with other components: The only reference that we had for $I_p$ reports the sum of $I_p$ and three background channels, which are different from the background channel used in the Inada model [64]. Also, $[Ca^{2+}]_i$ was only reported in the context of the full cell model by Inada *et al.*

While our simulations are mostly in qualitative agreement with the reference plots, we could not always achieve an exact match. We assume that this is due to further unreported changes in parameter values. For example, for the current density time course of $I_{st}$ in S5B Fig we had to set the parameter $g_{st}$ to 0.27 nS instead of 0.1 nS as reported in [7]; the differences for $I_{NaCa}$ in S6A and S6B Figs vanish when the current densities are multiplied by a scaling factor 1.18, which can be achieved by adjusting $k_{NaCa}$ accordingly; and for $I_{to}$ the current in S2E Fig is slightly lower than in our model, which could be explained if $T_{hold}$ was too small to allow a full recovery to the steady state in [7]. Finally, the differences in the full cell models might be explained if $I_{ACh}$ *was* actually used for simulations. There are only two instances of qualitative differences for $I_f$ in S4C Fig, and for $I_{st}$ in S5C Fig. We have no good explanations for these

differences, but it is unlikely that they are due to an error on our side, since they exist only in the I-V-plots, but not in other plots using the same data or reference plots from other sources.

**3.5.1 Solution 1: Run experiments in continuous integration.** The hurdles to results reproducibility posed by missing and erroneous information about reference plots and by missing plots themselves can also be solved by employing automated testing. The test cases in InaMo directly produce the simulation data required for a specific reference plot. The model code contains the full experiment protocol including all relevant simulation settings such as the solver and the step size. An example can be seen in Listing 7. The repository also contains a plotting script that reads the simulation output produced by the test script and generates plots for all examples. In consequence, reproduction of the methods of this article becomes possible with a few simple steps: Researchers have to install OpenModelica, the Python distribution Anaconda, and Julia with the single additional package `ModelicaScriptingTools`. Then they can download our code from GitHub and type the following two commands in a command prompt:

```
julia --project="." scripts/unittests.jl
python scripts/plot_validation.py
```

This should result in the creation of a directory called `plots` which contains a reproduction of all the reference plots listed in Table 4. Only $I_{ACh}$ remains untested in InaMo, because we do not have any reference for the equations used in the C++ code of Inada *et al*.

**3.5.2 Solution 2: Use dummy components in unit tests.** While we did not have a reference plot for the $[Ca^{2+}]$ handling, we still wanted to create a unit test of the component as it was quite difficult to implement, and we wanted to isolate it from any feedback loops to facilitate bug fixing. In software engineering, it is a common issue that a piece of code that should be tested depends on a fairly complex and not fully predictable environment, such as a database or a web service. In these cases, dummy components are used, which provide the same interface as the required service, but actually contain no logic whatsoever and only return the results that are expected and needed for the unit test.

This technique can also be applied to mathematical modeling. For the unit test of the $[Ca^{2+}]$ handling, we approximated the time course of the currents $I_{Ca,L}$, and $I_{NaCa}$ throughout an action potential in the full cell example very roughly with a sum of Gaussians. This leaves us with current signals that have a physiologically plausible shape and value and that do not depend on any other component. The resulting plot therefore shows the behavior of the $[Ca^{2+}]$ handling component in isolation, allowing to examine the effect of changes to this component in a controlled environment.

**3.5.3 Solution 3: Publish simulation data used for regression tests.** Since we only had plots as a reference, we initially only checked the exactness of our experiment results by comparing the plotted values at prominent sample points like extrema or zero crossings. For the full cell model, we invested the additional effort to reconstruct the simulation data from the plots using the vector graphics editor Inkscape and a small Python script. We then later extended this reconstruction procedure to all other reference plots. This allowed us to immediately assess whether a parameter change brought the simulation result closer to the original data or introduced additional deviations. However, this process is both tedious and inexact. In a first attempt, we underestimated the scale of the x-axis in the plot for the full cell model, which was only given as a small ruler-like segment of 50 milliseconds width. Additionally, we first assumed that the test pulse occurred exactly after 50 milliseconds for each cell type, but later found out that the position differed by a few milliseconds between plots. These errors and the reconstruction effort could have been eliminated, if the simulation data used for the original plots was available for download.

As mentioned above, we publish our simulation output for the regression tests, which includes all data required to reconstruct our reference plots. We also make the reconstructed simulation data from the plots in the original article available. Additionally, our plotting script can be easily configured to produce plots from the reference data instead of or in addition to the simulation output. Therefore, even if there should be some unforeseen issues with running one of our scripts in the future, an exact reproduction of the simulation results will still be possible, because the reference data allows to reliably quantify the error in a reproduction attempt.

## 3.6 Semantics lost in the chain of references

**3.6.1 Problem description.** The last problem that we encountered in our reproduction of the results of the Inada model was not so much concerned with correctness and completeness but with the understandability of the model. In an attempt to reproduce simulation results, it is unlikely that the goal is to reproduce the full code with the exact same structure as before. This was also the case for us, as we wanted to include the model in a high-level model of the human baroreflex [66, 67]. For this task, we also wanted to adhere to our MoDROGH guidelines [22]. This required us, among other changes, to bring the model into a modular structure that follows the biological structure as much as possible. For the HH-type ion channel formulations this was straightforward, although we sometimes struggled to understand the reasoning behind the choice of fitting functions.

The $I_{Na}$ formulation, however, follows the Goldman-Hodgkin-Katz (GHK) flux equation, which—unless one is already familiar with this equation—only becomes apparent when reading the reference by Lindblad *et al.* [65]. This posed a problem, because understanding this equation was required for resolving an error in the article: The permeability $P_{Na}$ was given in nl/s by Lindblad *et al.*, which is not a unit for permeability and also has the wrong order of magnitude since it should be pl/(s · m$^2$). This error can only be found and fixed, if one understands the semantics that $P_{Na}$ is supposed to be the permeability term used in the GHK flux equation.

A similar but more severe problem occurred in the formulation for $I_{NaCa}$, where the main set of equations that defines the current is the following:

$$x_1 = k_{34}k_{41}(k_{23} + k_{21}) + k_{21}k_{32}(k_{43} + k_{41}) \tag{1}$$

$$x_2 = k_{43}k_{32}(k_{14} + k_{12}) + k_{41}k_{12}(k_{34} + k_{32}) \tag{2}$$

$$x_3 = k_{43}k_{14}(k_{23} + k_{21}) + k_{12}k_{23}(k_{43} + k_{41}) \tag{3}$$

$$x_4 = k_{34}k_{23}(k_{14} + k_{12}) + k_{21}k_{14}(k_{34} + k_{32}) \tag{4}$$

$$I_{NaCa} = k_{NaCa}(k_{21}x_2 - k_{12}x_1)/(x_1 + x_2 + x_3 + x_4); \tag{5}$$

Without further explanation it is nearly impossible to see that this is an analytic solution to the diffusion equations between four states of the sodium potassium pump, of which only the state transitions between state 1 and 2 are electrogenic. Inada *et al.* cite Kurata *et al.* as direct source for $I_{NaCa}$, but to obtain an explanation of the rationale behind the equations, one has to go one step further to an article by Matsuoka *et al.* [57]. This information was important for us since it meant that we could not further modularize $I_{NaCa}$, because it would not have been possible to automatically extract the analytic solution from individual diffusion models.

The last and most important example of lost semantics was the $[Ca^{2+}]$ handling. Here, the equations describe the transport of $Ca^{2+}$ cations between four compartments. This is not

$$\frac{\mathrm{d}\left[Ca^{2+}\right]_{\mathrm{sub}}}{\mathrm{d}t} = \frac{-I_{\mathrm{Ca,L}}}{2FV_{\mathrm{sub}}} \quad (a)$$

$$+ \frac{2I_{\mathrm{NaCa}}}{2FV_{\mathrm{sub}}} \quad (b)$$

$$+ j_{\mathrm{rel}}\frac{V_{\mathrm{rel}}}{V_{\mathrm{sub}}} \quad (c)$$

$$- j_{\mathrm{Ca,dif}} \quad (d)$$

$$- [CM]_{\mathrm{tot}}\frac{\mathrm{d}f_{\mathrm{CMs}}}{\mathrm{d}t} \quad (e)$$

$$- SL_{\mathrm{tot}}\frac{\mathrm{d}f_{\mathrm{CSL}}}{\mathrm{d}t} \quad (f)$$

**Fig 3. $[Ca^{2+}]$ handling in the Inada model.** Left: One of the 15 differential equations that govern the intracellular calcium concentrations as presented in the original article. This single equation mixes the following six physiological effects: the transport of calcium cations through the L-type calcium channels (a) and the sodium-calcium exchanger (b), the release of calcium from the JSR into the subspace via ryanodine receptors (c), the diffusion from the subspace into the cytosol (d), and the calcium buffer calmodulin in the subspace (e) and in the sarcolemma (f). Right: Graphical representation of the $[Ca^{2+}]$ handling in InaMo version 1.4.1. Each component represents a single physiological effect or quantity with intuitive icons for concentrations (beaker), calcium buffers (stylized protein), diffusion reactions (arrow from high to low concentration of circles), the ryanodine receptor (pore in lipid bilayer), and the SERCA pump (scissor-like structure in lipid bilayer). Effects (c)–(f) of the left-hand side equation are represented by the four components connected to the beaker on the upper left (marked in red), while effects (a) and (b) are handled by the external ion channel components when they are connected to the large calcium connector (blue circle) on the center left. These external connections can be seen in Fig 6 (left).

apparent in [7], but only in [54], which contains a graphical representation of the model. However, Kurata *et al.* still do not couple this understandable graphical representation with the actual equations. Instead of separating them into diffusion reactions, the ryanodine receptor and the SERCA pump, they are only grouped by compartments. Additionally, the effects of all ionic currents on the $Ca^{2+}$ concentration are lumped together in the same equations, which further complicates understanding. An illustration of this problem can be seen in Fig 3. Even after disentangling the equations into small components, we were still confused by the volume terms that were applied to the "flux" variables $j_{\mathrm{rel}}$, $j_{\mathrm{up}}$, $j_{\mathrm{tr}}$, and $j_{\mathrm{Ca,diff}}$ in seemingly arbitrary fashion. An example can be seen in Fig 4. The reason behind this confusing use of volume terms is that the original equations only use *concentrations*, but the transport has to conserve the

General rule:

$$\frac{\mathrm{d}\left[Ca^{2+}\right]_{\mathrm{src}}}{\mathrm{d}t} = \ldots - \frac{\min(V_{\mathrm{src}},V_{\mathrm{dst}})}{V_{\mathrm{src}}}j_{\mathrm{src,dst}} + \ldots$$

$$\frac{\mathrm{d}\left[Ca^{2+}\right]_{\mathrm{dst}}}{\mathrm{d}t} = \ldots + \frac{\min(V_{\mathrm{src}},V_{\mathrm{dst}})}{V_{\mathrm{dst}}}j_{\mathrm{src,dst}} + \ldots$$

Equations in article:

$$\frac{\mathrm{d}\left[Ca^{2+}\right]_{\mathrm{up}}}{\mathrm{d}t} = \ldots - j_{\mathrm{tr}}\frac{V_{\mathrm{rel}}}{V_{\mathrm{up}}} + \ldots$$

$$\frac{\mathrm{d}\left[Ca^{2+}\right]_{\mathrm{rel}}}{\mathrm{d}t} = \ldots + j_{\mathrm{tr}} + \ldots$$

**Fig 4. Comparison between general rule for inactive transport equations (left) and actual equations occurring in the article by Inada *et al.* (right).** The right-hand side is the result of substituting src = up and dst = rel in the left-hand side and then simplifying due to $\min(V_{\mathrm{up}},V_{\mathrm{rel}}) = V_{\mathrm{rel}}$. If only the right-hand side is given, it is not trivial to trace back these steps to arrive at the general rule, which is required to understand the meaning of the equation. The name "tr" does not immediately make it apparent what are the source (src) and destination (dst) concentrations affected by $j_{\mathrm{tr}}$ and since $V_{\mathrm{rel}}/V_{\mathrm{rel}}$ cancels out in the second equation, the structure is also lost.

$$h_{1\infty} = \ldots$$

$$\tau_{h_1} = \frac{0.03}{1 + \exp\left((V + 40)/6\right)} + 0.00035$$

$$\frac{\mathrm{d}h_1}{\mathrm{d}t} = \frac{h_{1\infty} - h_1}{\tau_{h_1}}$$

```
GateTS inact_fast(
  redeclare function fsteady = ...,
  redeclare function ftau =
    genLogistic(
      y_min=0.00035, y_max=0.03+0.00035,
      x0=-0.040, sx=-1000/6.0
    )
) "inactivation gate (type1/h1)";
```

**Fig 5. Equations for the fast inactivation gate of $I_{Na}$ in the original article (left) and in InaMo (right).** The equations on the left-hand side constitute a typical description of an HH-type ion channel using a steady state $h_{1\infty}$ and a time constant $\tau_{h_1}$ to define the time course of the gating variable $h_1$ via a differential equation. The equation for $\tau_{h_1}$ was found by fitting a generalized logistic function to experimental data. It has a declining sigmoid shape with an inflection point at 40 mV, a minimum of 0.35 ms, and a maximum of 30.35 ms. This may be apparent for an expert modeler, who is familiar with similar models, but not to novices or biologists without a deep mathematical background. The InaMo code on the right-hand side therefore aims to make this expert view of the equations available to non-experts by capturing common equation structures in named and documented components. `GateTS` defines a HH-type gating variable based on the two replaceable functions `fsteady` for the steady state and `ftau` for the time constant. `genLogistic` is a fitting function, whose parameters are explained in its documentation: `y_min` is the minimum, `y_max` is the maximum, `x0` is the inflection point, and `sx` determines the steepness and direction (`sx` < 0 yields a declining sigmoid shape).

*amount of substance* between both sides. This introduces the need to convert from concentrations to amounts of substance (by multiplying with a volume term) and then back to concentrations (by dividing by another volume term). Even with this background knowledge, which is assumed by the articles about the $[Ca^{2+}]$ handling, it is not trivial to infer the general rule from the equations in the article. This is due to simplifications, which obscure the common equation structure, and the naming of the flux variables, which does not clearly indicate source and destination of the corresponding transport effect.

**3.6.2 Solution 1: Model design utilizing MoDROGH criteria.** The software engineering equivalent of these unclear, entangled and undocumented model semantics is termed "spaghetti code", which is code that is hard to maintain, because the program flow is hard to follow. The solution to this problem is a combination of modularization, documentation and clear design patterns for the code. As mentioned in Section 2.4, InaMo follows the guidelines associated for building models with a language that is modular, descriptive, human-readable, open, graphical, and hybrid (MoDROGH), which can increase the understandability as well as the methods and results reproducibility of a model [22].

The issue with non-transparent fitting functions is solved by defining a set of fitting functions with understandable names and a common structure for gate components. As Fig 5 shows, this allows to understand the gate equations without having to untangle the structure of the fitting functions in memory. For example, the most common fitting function `genLogistic` can be quickly identified as a sigmoid function, whose parameter `x0` defines the point of maximum steepness, while `y_min` and `y_max` define the minimum and maximum value that the function can assume. It also becomes apparent that almost all gates use HH-type equations governed by a time constant and a steady state function.

Similarly, the GHK flux equation is implemented in a separate component that features both a detailed documentation in HTML format and explicit unit definitions, including the custom type `PermeabilityFM` that is used to document the unusual unit used for $P_{Na}$. This also fixes a minor issue mentioned in Section 3.1, as the documentation also explains the handling of the avoidable discontinuity in the function.

As mentioned above, the equations for the sodium calcium exchanger unfortunately could not be modularized to make more explicit that they are intended to model diffusion reactions between four states. However, we added a documentation string to each variable explaining its physiological interpretation. We also added the variables $E_1$–$E_4$ from Matsuoka *et al.*, which

represent the actual ratio of molecules in each of the four states and therefore facilitate the interpretation of the behavior of the component.

Finally, we already showed the effect that modularization has on the $[Ca^{2+}]$ handling in Fig 3. By separating the model into modules that each only represent a single physiological effect, these individual effects become more understandable. Readers can focus on understanding one module at a time, grouping the equations and parameters in memory to form a concept that can easily be recalled. With these concepts in mind, understanding the whole component becomes possible by inspecting its graphical representation, which shows how the individual effects are connected. This is facilitated by the fact that the graphical representation is neither a separate biological drawing, nor an automatically generated graph, but rather an accurate representation of the model defined with Modelica constructs similar to a circuit diagram. An example showing the definition of the graphical representation in the Modelica code can be seen in Fig 6. On the code and equation level, InaMo uses amounts of substance instead of concentrations as interface. This leads to a more natural representation of active and inactive transport components, which explicitly ensure conservation of mass. The diffusion reactions and the ryanodine receptor use a common base class `InactiveChemicalTransport`, which clearly explains the use of volume terms and presents the gradient-based transport equations in their general, more understandable form. Additionally, we change the naming of the individual concentrations from $[Ca^{2+}]_i$, $[Ca^{2+}]_{up}$ and $[Ca^{2+}]_{rel}$ to $[Ca^{2+}]_{cyto}$, $[Ca^{2+}]_{nsr}$ and $[Ca^{2+}]_{jsr}$ respectively, which allows us to also assign intuitive names to the transport

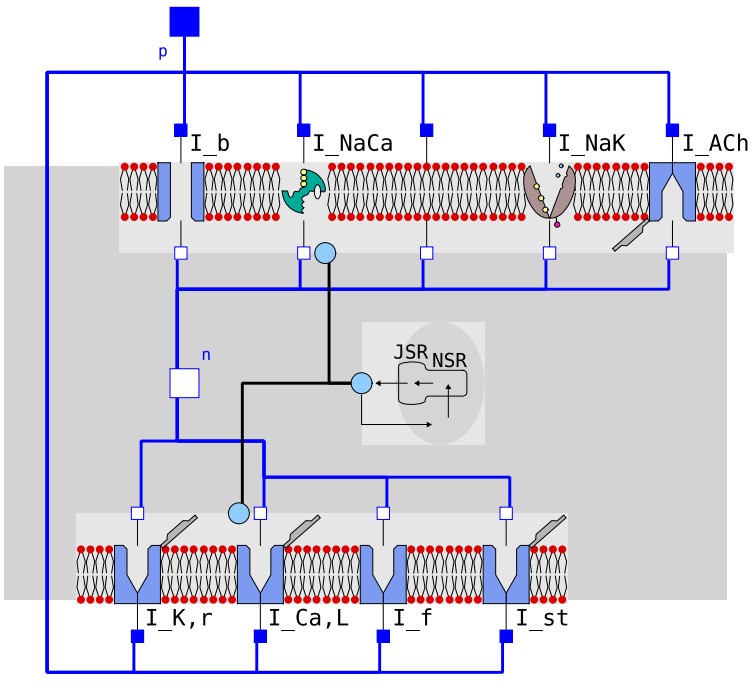

```
model NCell
  BackgroundChannel bg
    annotation(Placement(
      transformation(
        origin = {-51, 53},
        extent = {{-17, -17}, {17, 17}}
      )
    ));
  ...
equation
  connect(bg.p, p) annotation(Line(
    points = {{-50, 70}, {-50, 100}},
    color = {0, 0, 255}
  ));
  ...
annotation(Diagram(graphics={Rectangle(
  fillColor = {211, 211, 211},
  pattern = LinePattern.None,
  fillPattern = FillPattern.Solid,
  extent = {{-100, 60}, {100, -60}}
)}));
end NCell;
```

**Fig 6. Graphical representation of the N cell model.** Left: Diagram resulting from drag and drop composition of model components (InaMo version 1.4.1). Right: Automatically generated embedding of graphical annotations in the model code showing the placement of the background channel (`annotation(Placement(...))`), the connection line to the positive pin (`annotation(Line(...))`) and the definition of the gray rectangle in the background (`annotation(Diagram(...))`).

components. For example, the module for the diffusion from the subspace to the cytosol is called `sub_cyto`.

Apart from these individual examples, InaMo uses the general MoDROGH guidelines to ensure that the model code reflects the physiological semantics as much as possible, making them transparent for the user. For example, the whole model only uses two kinds of interfaces: an electrical interface for ion currents and a chemical interface governing the changes in the amount of ions in a compartment. Following the convention that outward currents are positive, each ionic current has a positive pin on the extracellular side and a negative pin on the intracellular side. The electrical interface is also compatible to electrical components of the Modelica standard library `Modelica.Electrical`, allowing standard electrical components such as a ground or current source to be used in InaMo. Both kinds of interfaces can be seen in Fig 6, where electrical connections are represented by blue squares, which are filled for positive pins, and chemical connections are represented by blue circles. It can also be seen that no component has more than three of these connections, which keeps the cognitive effort required to understand them at a low level.

**Listing 9**. Interface for electrical connections between model components in the Modelica standard library. The keyword `flow` establishes an acausal connection with the conservation law that the sum of the `i` variables of all connected components must be zero.

```
connector PositivePin "Positive pin of an electrical component"
  SI.ElectricPotential v "Potential at the pin" annotation(...);
  flow SI.Current i "Current flowing into the pin" annotation (...);
  annotation (...);
end PositivePin;
```

One important aspect of these interfaces is that they are *acausal*, which means that no prior assumption is made which variables are defined by input signals and which will be observed as the output of an experiment. For example, this means the same model code can be used for voltage- and current-clamp experiments. Modelica achieves this by using connector variables with the `flow` keyword and automatically generating conservation law equations representing Kirchhoff's current law for the electrical interfaces and the conservation of mass for the chemical interface. An example of such a connector definition can be seen in Listing 9. The most important effect of this feature for the design of the model is that adding or removing ion currents is as easy as adding and removing the component and its connections in the graphical representation, which automatically adds or removes the required term to the conservation law equations.

These two physical connectors ensure that the model structure in the code follows the biological structure of the modeled system. The full cell is composed of models of the lipid bilayer, ion channels, ionic pumps, and the $[Ca^{2+}]$ handling, which only exposes $[Ca^{2+}]_{sub}$ as the concentration that is relevant for the ion currents. The ion channels, in turn, contain gate models which are composed of basic HH-type gates with predefined or custom fitting functions. This structure closely ties the equations to their semantic meaning and therefore facilitates interpretation.

At the lowest hierarchical level, each variable and parameter in the model is annotated with proper units following the SI and has both a human-readable name and a documentation string explaining its physiological role. Models that are more involved additionally contain a documentation text in HTML format with detailed information about the model structure. This ensures that the model is understandable without further literature research.

**3.6.3 Solution 2: Annotation of sources and rationale for parameter values.** To spare researchers that want to reproduce our model skimming through a large body of literature as in Fig 1 and to make our parameter choices transparent, we annotated the experiments with a

literature source or rationale for each parameter value. Currently, this is done within the HTML documentation string of the models defining the experiments. However, Modelica also allows using so-called vendor-specific annotations to add structured annotations with custom content. This feature could be used to make these annotations not only human- but also machine-readable and, for example, allow to automatically add this information to the table of parameters generated by MoST.jl.

## 4 Discussion

### 4.1 Answer to RQ1

In research question RQ1, we asked which factors hindered the reproduction of the methods and results of the Inada model. Despite the efforts of the authors to provide detailed reference plots and publish their code, a considerable reverse engineering effort was required to build our Modelica implementation InaMo. Fig 1 in S1 Text shows an estimation of the distribution of the working time that went into InaMo. In total, the development took us an estimated 86 work days (i.e. four months). Small errors in published equations and parameter values required the debugging (19 working days) of individual parts of the model. This debugging was hindered by missing information about some model components and missing and incomplete reference code and experiment protocols. This in turn required further literature research (33 working days) to recapture the model components and the semantics of their equations, which revealed several issues with the understandability of the equations, which were structured for ease of implementation and not for ease of understanding. It is quite conceivable that other researchers before us have encountered these issues and decided that the benefit of including the Inada model in their research did not warrant the effort that would be required to do so. This is especially unfortunate since it is a ground-breaking model with several interesting properties, which deserves more attention.

### 4.2 Answer to RQ2

In our second research question RQ2, we wanted to know what can be done to overcome or to avoid these reproducibility issues. With InaMo we have adopted a model engineering strategy, that uses a suitable MoDROGH language to apply proved techniques from software engineering to the problem. These approaches broadly fall into three categories: First, we established an automated testing pipeline using CI and CD techniques to guarantee completeness and methods reproducibility of the published version of the model. This also includes automated unit consistency checks, performing the actual simulations used for plots in the article in the CI pipeline, and publishing the simulation data both for the reproduction of results and to use them in regression tests. Second, we paid special attention to the model design to increase the understandability and reusability of the model code, using a MoDROGH language and building a component hierarchy with two simple interfaces and small independent components, which only represent a single physiological effect or compartment, and which can be combined via drag and drop in an easily interpretable graphical representation. Third, we provided extensive documentation both within the model in the form of unit definitions, human-readable variable names, and embedded HTML documents and through external services for version control and archival.

Both testing (11 working days) and refactoring and documentation (16 working days) took considerable effort (see Fig 1 in S1 Text). However, we found that this additional effort was well justified by the benefits gained during development, even if we do not consider the benefits for other researchers who want to reproduce our methods or results. For debugging, it was invaluable to have a CI pipeline performing regression tests for individual components,

because we would immediately notice when a change accidentally introduced an error in other models than the one that was currently under development. Utilizing the version control system, we could quickly identify and roll back the changes that introduced bugs. As concrete example, we added unit and regression tests for the individual components of the [$Ca^{2+}$] handling precisely because they helped us to ensure that our transformation of the model structure from a concentration interface to an interface using amounts of substance did not change the simulation output. Transforming the model into a modular structure that follows the biological structure of the modeled system also helped us to notice some of the errors in the original model, which we would otherwise have overlooked. Much like explaining a concept to somebody else can reveal own misconceptions, making a model more understandable can reveal potential error sources. Additionally, documenting the meaning of variables and parameters as well as the source and rationale of parameter values meant that we only had to look up such information once. This reduced the time required to get an overview over a part of the code that we had to revisit a few weeks or months after it was written. We are positive that without these measures, finding the last errors that prevented us from obtaining reasonable simulation results with the full cell models would have taken considerably more time, if we had achieved a working reproduction at all.

It also has to be noted that the effort required for the solutions presented in this article can and already has been reduced for other researchers. During the development of InaMo, we created the Julia library MoST.jl, which allows setting up tests with only three relevant lines of code (see Listing 5) and provides more and better readable error messages than the OpenModelica compiler when used with default settings. Setting up these tests on a CI service does not require much more effort. In fact, if the same folder structure is used as in our project, it would be possible to simply copy the GitHub Actions configuration script shown in Listing 1.

Furthermore, the systems biology community could choose to provide own CI/CD pipelines using open source tools like Jenkins [68, 69] or services like NF-CORE [25] or FAIR-DOMHub [70], which are already established in bioinformatics and systems biology. This way, specific virtual machine images and/or pipelines for common modeling languages could be provided, which already include all necessary tools for simulation and plotting. This would further reduce both the size of the setup script and the execution time required.

## 4.3 Generalization and alternatives

While this work is only a case study of the Inada model, we believe that the issues that we found here and the solutions that we presented can be highly relevant for mathematical modeling in systems biology in general. For example, the aforementioned reproducibility study of models in the BioModels database found very similar errors and reproducibility hurdles in half of the 455 examined models [27]. In summary, this study lists the following reproducibility issues: sign errors, missing terms in equations, typing errors in parameter values, unit errors, missing or incorrect parameter values, missing or incorrect initial concentrations, errors in equations, ambiguous or inconsistent variable names, and poor readability and lacking documentation in the code. Our case study of the Inada model showcased concrete examples for each of these categories, which indicates that it at least can be representative for these 455 other models. If the issues are similar, it is reasonable to assume that the same or similar solutions like the ones that we used here will also work for other models.

This is further backed by the fact that the software engineering techniques that we applied, such as version control, CI and delivery, automated testing, modularization, and documentation, are not limited to any specific property of the Inada model. They can be, and in fact are, applied to all kinds of software solutions. There are some adjustments required for

mathematical modeling, such as the development of specialized testing libraries like MoST.jl. However, there is no reason to believe that there is any area of mathematical modeling that cannot benefit from these general techniques in some way.

We also think that the Inada model is a fitting example to represent reproducibility challenges in the development of multi-scale models, including a large number of equations, the combination of different preexisting models, and the need to incorporate the model into a larger multi-scale context. As mentioned in Section 1, the three groups that did reproduce the results of Inada *et al.*, did so in a multi-scale context, and this was also our original purpose. The model is in itself a combination of multiple existing models for ionic currents and the $[Ca^{2+}]$ handling by the sarcoplasmic reticulum. With its 116 equations, 79 parameters, and 27 initial values, which are only partly shared between the three different cell types, it is large enough that it can no longer be handled well in a classic monolithic structure that only lists all equations in a loosely grouped fashion. The Inada model therefore shows that even a well-crafted and relevant model can be subject to reproducibility issues, simply because of its inherent complexity.

Furthermore, our findings are not specific to the language Modelica. Integration for scripting languages like Python or Julia also exist, for example, for SBML [71, 72] and CellML [73]. This is sufficient to utilize the unit testing features of these languages and to build a simulation script that can be run in a CI pipeline. One remaining caveat is the need to download and install all software necessary to run the script on the CI server, which rules out proprietary solutions with expensive licenses such as MATLAB/Simscape. Regarding the model design utilizing MoDROGH criteria, our previous work shows that multiple languages exhibit MoDROGH criteria [22] and illustrates trade-offs between different choices. We did use some Modelica features for our design that do not exist in other languages like SBML and CellML. This includes the graphical composition of models, object-oriented programming with multiple inheritance, acausal connections between electrical and chemical components, the grouping of interface variables to connectors, and the annotation of the experiment setup within the model file itself. It is also interesting to note that unlike CellML and SBML, which are mainly designed as exchange formats, Modelica code focuses on human-readability over machine-readability and is designed to be directly written by humans. This removes tool-specific barriers between model designer and model user and avoids clutter in version control systems [22]. However, Modelica also has downsides: A CI/CD pipeline is only possible with the open source compiler from OpenModelica, and not with the proprietary compiler for the IDE Dymola, which is more widespread in industry and not fully compatible with OpenModelica, although both implement the same language standard. Additionally, Modelica is a general purpose modeling language, which lacks biology-specific features and language constructs like annotation of components with ontology terms, or the `<kineticLaw>` tag in SBML.

As an important implication, Modelica and SBML or CellML tools are not interoperable. This is important, because interoperability allows model users to reproduce results using the tools that they are familiar with and thus to easily combine models. Modelica's mechanism for interoperability is the FMI that allows to create executable artifacts, so called FMU, from models, which can be reused even across different languages. On the downside, these FMUs are mostly opaque boxes. They can contain source files in C and a list of variables and equations in JavaScript Object Notation (JSON), but they are not suitable for results replication with modifications that go beyond changing parameter settings. In contrast, SBML and CellML are directly designed as exchange formats, which is why they are based on XML. Both FMI and SBML report support by over 100 tools [63, 74], but crossing between the two ecosystems is more difficult. SBML2Modelica allows to directly translate SBML models to Modelica [75],

but we are not aware of any tool that operates in the opposite direction. The only tools used in systems biology that also support FMI currently are MATLAB with the SimBiology and the FMI toolbox, and custom Python code using appropriate libraries for both standards. As a first remedy, FMI support could be added in SBML and CellML tools. Alternatively, the systems biology community could, like Modelica, adopt the software engineering practice to distinguish between "source" and "distribution" formats for models. In this analogy, SBML and CellML would be used as distribution formats, which are used to make models easily accessible for simulations by other researchers, but models would additionally be published in a more human-readable and version control-friendly source language like Antimony [40] or CellML Text [53], which can directly be used for model development. In the case of this article, it would be ideal to have a Modelica2SBML tool, that compiles from the source language Modelica to the distribution language SBML. This is not possible in general, because Modelica supports more formalisms than SBML, but it might be possible for a subset of the language. As a first compilation step, Modelica models are transformed into a "flat" model that collects all variables, functions, events, and equations in a single file without any hierarchy or modular structure. If the translation process is restricted to a subset of the Modelica language, a translation of a "flat" model to SBML code might be achievable. However, this would also mean that the benefit of the modularity and understandability of Modelica models is largely lost in translation. It becomes clear that further research is needed to bridge this gap.

Regarding alternatives, GitHub Actions is not the only choice to implement a CI/CD pipeline. The open source project Jenkins [68, 69] can be used to set up a server that is controlled by a scientific institution, a journal, or a specific lab, alleviating privacy concerns when working with patient data. Additionally, other major open source repository hosting providers like BitBucket [76] and GitLab [77] also offer CI pipelines with varying amounts of free computing time for open source projects. Finally, modeling-specific solutions could be implemented in existing workflow environments like NF-CORE [25] or FAIRDOMHub [70].

Our findings can also be seen in a more general light with respect to the FAIR Guiding Principles for scientific data management and stewardship [78], since they contribute to making the model code findable, accessible, interoperable, and reproducible (FAIR). InaMo is findable in the Zenodo database [38], on GitHub (https://github.com/CSchoel/inamo), and in the BioModels database (https://www.ebi.ac.uk/biomodels/MODEL2102090002). Zenodo allows us to cite specific versions of the code with a unique DOI and GitHub provides a platform for discussing issues and open questions regarding the implementation. BioModels, Zenodo, and the GitHub Archive Program also contribute to making the code accessible for future researchers. Interoperability is provided by the CI/CD pipeline, which ensures that the code runs on other machines. Additionally, the main model is exported as executable artifact using the FMI, which allows to incorporate it in other projects even across different modeling languages. The modular design utilizing MoDROGH criteria and the additional documentation effort for InaMo do not only facilitate reproduction but also reuse, because the model becomes more understandable and extensible. Additionally, each published version of the model uses an open license (MIT license for Zenodo and GitHub, and Creative Commons CC0 1.0 for BioModels). However, as mentioned before, Modelica does not directly support the annotation of model parts with ontology terms. For full compliance with the FAIR Guiding Principles, this has to be addressed either by using vendor-specific annotations and developing tools that can read and write ontology data in Modelica models, or by implementing common ontologies like the systems biology ontology (SBO) [79] as a type hierarchy in Modelica as outlined in [22].

### 4.4 Limitations

There are some limitations to our approach regarding unit testing. First, unit tests are only meaningful, if the "unit" in question can be used in a simulation that does not involve other complex components. For example, for the $[Ca^{2+}]$ handling we had to create dummy components to mimic the time course of $I_{Ca,L}$ and $I_{NaCa}$ during an action potential in order to obtain a meaningful curve for $[Ca^{2+}]_i$. It is possible that other models may include components that require so many connections to other parts of the model that creating a unit test requires a lot of effort. However, it can be argued that such a component should then be investigated for possible design flaws, since the goal in a modular design is to minimize the interface of a component.

Another problem can be the lack of reference data. Our current unit tests already can be criticized, because they do not follow the usual pattern of a test that has a defined input and an expected output. We only test that the simulations runs error-free and the output is only compared to the output of a previous iteration with the help of regression tests. Tying this output to the actual goal of approximating measured data from biological experiments is currently still performed by a human who has to inspect and compare the resulting plots. For InaMo, this was the only approach we could take, since most of the experimental data was only available as plots, and we would have to reconstruct the original data points by hand. We did this for the simulation output of the models, but not for the experimental data, because the latter is even harder to read from the plots. If the data were published and included in the repository, it would also be possible to define a new kind of test in MoST.jl, which tests the agreement between experimental data and simulation output by some metric such as the root-mean-square error. However, this is of course only possible if such data is available and this may not be the case at every level of detail, limiting the usefulness of unit testing.

Additionally, it has to be considered that test suites like ours can become computationally demanding. We currently run the full simulations that we use to produce our result plots from within the test suite, because it is convenient to only need one script and because this ensures that the CI server reproduces our methods in every iteration. However, if we had included the full one-dimensional model by Inada *et al.* with hundreds of cells, this would mean that our test suite might not run in a few minutes but instead require hours. This prolongs the response time unduly, which limits the usability for quickly testing and debugging small changes to the models. One solution to this problem is to limit the length or size of the simulations in the test suite and to add a second script that is used to produce the actual simulation result. However, this then introduces a source for errors since the content of this new script cannot be tested using CI. For example, in another model, we encountered an error related to the synchronization between two event sources that only occurred after 170 seconds of simulated time.

Apart from the computational effort, the human effort in designing a model with MoD-ROGH criteria can also be significant. Most modelers probably did not receive training in software engineering and therefore first have to learn to apply design patterns. This is especially difficult, because while guidelines can help, good software design ultimately arises from experience and experimentation. We argue that the benefits are worth the additional effort, but the initial barrier may be high for many systems biologists.

Even if an understandable modular model structure is achieved during development, it is still likely that the model has to be translated to a grouped list of equations for presentation in a scientific article or even just to communicate some details to a researcher who is more familiar with this representation. Even though OpenModelica does allow to inspect a flattened version of arbitrarily complex models, this representation includes a lot of visual clutter due to alias variables that are introduced by the hierarchical structure. It is therefore not trivial to

translate this code into a human-readable list of equations. This task can be facilitated by libraries producing automated documentation like MoST.jl, but this feature of MoST.jl is still in an experimental stage. At the same time, we are not aware of any other approaches that provide similar facilities for flattening highly modular model structures in an equation-based format while retaining the grouping information from the modular design.

A similar argument can be made about documentation. In software engineering, there is little doubt that documentation is valuable and even essential for understandable and maintainable code, yet it is often lacking, even in large, successful projects. This is because writing good documentation requires a lot of time and does not generate its benefit at the time of writing, but only at a later stage in the project. For systems biology in particular, we see the concern that there is not much incentive to document code beyond ones own understanding. This would be different, if academic journals did not only require the code to be available but also had some requirements for understandability and documentation standards.

## 5 Conclusion

From our case study we can derive several suggestions for tackling reproducibility issues in mathematical modeling in systems biology. Using a CI service, like GitHub Actions, in conjunction with unit and regression tests that are as fine-grained as possible can guarantee methods reproducibility and the completeness of the published code and data. The more automated tests can be performed within such a system, the better the chances for the model to be reproducible and reusable in different ways. It might be worthwhile for the systems biology community to consider implementing or using a CI service with predefined virtual machine images for typical modeling workflows. These images could then be archived allowing not only the long-term storage of the model code but also of the software that was used to simulate it. Journals like *PLOS Computational Biology* and the *Physiome Journal*, which already employ rigorous testing of reproducibility standards by reviewers, might be able to host such a service to provide authors with a standardized mechanism to facilitate reproducibility and to reduce the burden placed on reviewers. Beyond methods reproducibility, results reproducibility cannot be guaranteed by automated tests. They do increase the likelihood that a reproduction attempt is successful, but it might still be complicated by missing documentation or poor understandability of the code. Here, the MoDROGH guidelines or similar "style guides" for model code, can help to make models approachable and reusable for other researchers. However, the only thing that truly guarantees results reproducibility is and remains an actual validation study. We therefore suggest that more of these studies should be performed and published and that there should be some way to indicate that the results of a model have been successfully reproduced in model repositories. In general, we find the philosophy of model engineering, i.e. the application of software engineering techniques to mathematical modeling, very promising. We think that building models with more care to design and engineering aspects will both benefit the scientific impact of a model and scientific progress in systems biology as a whole. In particular, we hope that InaMo, our understandable implementation of the Inada model with reproducible methods and results, can kick-start some new projects on the electrophysiological properties of the AV node.

## Supporting information

**S1 Text. Data supplement.**
(PDF)

## Author Contributions

**Conceptualization:** Christopher Schölzel.

**Data curation:** Christopher Schölzel.

**Formal analysis:** Christopher Schölzel.

**Investigation:** Christopher Schölzel.

**Methodology:** Christopher Schölzel.

**Software:** Christopher Schölzel.

**Supervision:** Alexander Goesmann, Andreas Dominik.

**Validation:** Christopher Schölzel, Valeria Blesius, Gernot Ernst.

**Visualization:** Christopher Schölzel.

**Writing – original draft:** Christopher Schölzel.

**Writing – review & editing:** Valeria Blesius, Gernot Ernst, Alexander Goesmann, Andreas Dominik.

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
