## [Decision Letter · Decision Letter 0]

10 May 2021

PONE-D-21-08027

Countering reproducibility issues in mathematical models with software engineering techniques: A case study using a one-dimensional mathematical model of the atrioventricular node

PLOS ONE

Dear Dr. Scholzel,

Thank you for submitting your manuscript to PLOS ONE. After careful consideration, we feel that it has merit but does not quite fully meet PLOS ONE’s publication criteria as it currently stands. Therefore, we invite you to submit a revised version of the manuscript that addresses the points raised during the review process.

To be considered for publication further, we ask that each point raised by Reviewer 1 be clarified in the text.  For example, "Perhaps the authors can create a new paragraph in the discussion section that acknowledges the model exchangeability issue, considers the pros and cons of various approaches, and makes recommendations regarding how the field could approach a solution to this issue in the future." My impression of the manuscript is that it is thorough, meticulously-researched and of use to the field. However, the many expressions of frustration with the Inada model should be minimized throughout the text.

A rebuttal letter that responds to each point raised by the academic editor and reviewer. You should upload this letter as a separate file labeled 'Response to Reviewer'.A marked-up copy of your manuscript that highlights changes made to the original version. You should upload this as a separate file labeled 'Revised Manuscript with Track Changes'.An unmarked version of your revised paper without tracked changes. You should upload this as a separate file labeled 'Manuscript'.

If applicable, we recommend that you deposit your laboratory protocols in protocols.io to enhance the reproducibility of your results. Protocols.io assigns your protocol its own identifier (DOI) so that it can be cited independently in the future. For instructions see: http://journals.plos.org/plosone/s/submission-guidelines#loc-laboratory-protocols . Additionally, PLOS ONE offers an option for publishing peer-reviewed Lab Protocol articles, which describe protocols hosted on protocols.io. Read more information on sharing protocols at https://plos.org/protocols?utm_medium=editorial-email&utm_source=authorletters&utm_campaign=protocols .

We look forward to receiving your revised manuscript.

Kind regards,

Roger A. Bannister, PhD

Academic Editor

PLOS ONE

Journal Requirements:

3) Please include captions for your Supporting Information files at the end of your manuscript, and update any in-text citations to match accordingly. Please see our Supporting Information guidelines for more information: http://journals.plos.org/plosone/s/supporting-information.

Reviewers' comments:

Reviewer's Responses to Questions

**Comments to the Author**

1. Is the manuscript technically sound, and do the data support the conclusions?

Reviewer #1: Yes

2. Has the statistical analysis been performed appropriately and rigorously? 

Reviewer #1: N/A

3. Have the authors made all data underlying the findings in their manuscript fully available?

Reviewer #1: Yes

4. Is the manuscript presented in an intelligible fashion and written in standard English?

Reviewer #1: Yes

5. Review Comments to the Author

Reviewer #1: Reproducibility of computational models in biology is an important problem that has received a great deal of attention from researchers in the form of standards for model, data, and simulation exchange, and workflow management tools to help ensure that modeling studies can be made to be reproducible-by-design.

Many researchers in the field are aware of the reproducibility problem surrounding computational modeling, and individual investigators can often list general challenges and pitfalls to reproducibility, but Schölzel et al. present a very thorough, very detailed case study which precisely and clearly identifies the issues encountered in reproducing a complex physiological model. Studies such as this are rare, as a considerable amount of effort is often required to reproduce previously published models, but such studies are crucially important if computational modeling is to move towards a state of reproducible-by-design. Schölzel et al make explicit the somewhat vague notions that many in the field (including myself) ascribe to the difficulties in reproducing models, and allow future work to focus on reproducibility-related issues that have been confirmed as significant by actual experience.

In addition, Schölzel et al also provide a reimplementation of the subject model in the Modelica modeling language, which will greatly benefit future users of the model. On the whole, the manuscript is both a guide to those wishing to make improvements in future modeling tools, standards, and workflows, as well as an excellent example for teaching how to build a model to very high standards of reproducibility.

My only criticism of the study would be that the authors’ use of Modelica comes at the expense of more exchangeable formats such as CellML/SBML. The Modelica code provided with this submission is very complete, but the CellML and SBML formats are quite a bit more portable, and can be imported and exported by many tools, and the current study does not appear to have an accompanying CellML/SBML model.

The authors argue that CellML/SBML unsuited to first-hand model authoring due to their lack of human readability and problematic interaction with version control systems. For this particular study, I agree with this assessment. I believe that providing CellML/SBML code for this model would be against the DRY principle and create synchronization issues and hence barriers to robust reproducibility (and certainly, it would not be difficult to derive a CellML/SBML version of this model given the wealth of tests and documentation provided by the authors).

However, I would view it as a regression if exchangeable standards such as CellML/SBML were eventually replaced with less transferable counterparts. I think the authors can address these issues as discussion items, rather than make any implementation changes. I think an ideal solution would be to pursue an authoring medium, perhaps a meta-language, that retains the clarity and pragmatism of Modelica (for example) but can be interconverted to/from standards like CellML/SBML. Perhaps the authors can create a new paragraph in the discussion section that acknowledges the model exchangeability issue, considers the pros and cons of various approaches, and makes recommendations regarding how the field could approach a solution to this issue in the future. The authors already mention some of these issues around line 900. This section could be further expanded to offer recommendations and potential solutions for practical modeling languages and workflows that (I hope) attain all of the authors’ criteria as well as exchangeability and annotation support that standards were designed to address.

6. PLOS authors have the option to publish the peer review history of their article (what does this mean?). If published, this will include your full peer review and any attached files.

Reviewer #1: No

---

## [Author Response · Author response to Decision Letter 0]

17 Jun 2021

(Response copied from rebuttal letter)

Comment by reviewer #1:

My only criticism of the study would be that the authors’ use of Modelica comes at the expense of more exchangeable formats such as CellML/SBML. The Modelica code provided with this submission is very complete, but the CellML and SBML formats are quite a bit more portable, and can be imported and exported by many tools, and the current study does not appear to have an accompanying CellML/SBML model.

The authors argue that CellML/SBML unsuited to first-hand model authoring due to their lack of human readability and problematic interaction with version control systems. For this particular study, I agree with this assessment. I believe that providing CellML/SBML code for this model would be against the DRY principle and create synchronization issues and hence barriers to robust reproducibility (and certainly, it would not be difficult to derive a CellML/SBML version of this model given the wealth of tests and documentation provided by the authors).

However, I would view it as a regression if exchangeable standards such as CellML/SBML were eventually replaced with less transferable counterparts. I think the authors can address these issues as discussion items, rather than make any implementation changes. I think an ideal solution would be to pursue an authoring medium, perhaps a meta-language, that retains the clarity and pragmatism of Modelica (for example) but can be interconverted to/from standards like CellML/SBML. Perhaps the authors can create a new paragraph in the discussion section that acknowledges the model exchangeability issue, considers the pros and cons of various approaches, and makes recommendations regarding how the field could approach a solution to this issue in the future. The authors already mention some of these issues around line 900. This section could be further expanded to offer recommendations and potential solutions for practical modeling languages and workflows that (I hope) attain all of the authors’ criteria as well as exchangeability and annotation support that standards were designed to address.

Response:

-----------

We fully agree with the reviewer, and thank them for pointing out this oversight. While we do favor Modelica for model design, CellML/SBML have an undeniable advantage as exchange formats. In fact, we would be happy to provide a separate SBML or CellML version of the model, if there was a way to generate it automatically from Modelica code. As far as we are aware, such a conversion is currently only possible in the opposite direction with SBML2Modelica. This provides some interoperability between the two domains, but is insufficient to make our implementation directly accessible within an SBML- or CellML-based tool. The following approaches could remedy this in future projects:

- SBML/CellML tools could be extended to support the Functional Mock-up Interface (FMI), which is the designated exchange format in the Modelica domain. The only modeling tool we are aware of that supports both FMI and SBML is MATLAB with the SimBiology and FMI toolboxes. To facilitate this form of interoperability, we now also provide the three main model as Functional Mock-up Unit (FMU). The downside is that FMUs are opaque boxes, which do not allow a partial reproduction of results that requires a structural modification of the model.

- Like the reviewer said, a standard, human-readable text representation of CellML/SBML models with sufficient support for modularity would allow to apply the same techniques presented in this article to CellML/SBML models. Antimony and CellML Text come to mind as possible candidates.

- A full automatic translation of Modelica to SBML/CellML would probably be impossible, because SBML/CellML does not have full support for DAEs, discrete variables, and the object-oriented modeling style of Modelica. However, it could be achieved for a well-defined subset of the language. Upon compilation, Modelica models are "flattened", removing the hierarchical structure and collecting all variables and equations in one single file. Starting from this representation, a translation to SBML/CellML seems achievable, yet still quite complicated due to discrete events, Kirchhoff-like algebraic equations, and custom functions with imperative syntax. As an additional downside, the "flat" Modelica code loses most of the benefits regarding understandability and modularity of the model.

As suggested by the reviewer, we added a paragraph in section 4.3 that discusses this issue.

Comment by the editor:

My impression of the manuscript is that it is thorough, meticulously-researched and of use to the field. However, the many expressions of frustration with the Inada model should be minimized throughout the text.

Response:

-----------

We agree with the editor. We already tried to avoid emotional or judgmental language, but realized that we can do a better job to keep a factual tone throughout the text. In particular, we noticed that the reader can get the impression that we blame the authors and/or the reviewers of the Inada et al. study for the reproducibility issues. However, while we cannot deny that we felt frustration throughout the development, we believe that such issues are a natural result of high model complexity and that they can only be solved reliably with automated tool support. We tried to clarify this throughout the manuscript and also highlight that we do believe that Inada et al. were interested in making their methods as transparent as possible. In a way this makes the Inada model even more interesting as an example, because it shows that even high quality research is subject to the issues that we encountered.

---

## [Decision Letter · Decision Letter 1]

5 Jul 2021

Countering reproducibility issues in mathematical models with software engineering techniques: A case study using a one-dimensional mathematical model of the atrioventricular node

PONE-D-21-08027R1

Dear Dr. Scholzel,

We’re pleased to inform you that your manuscript has been judged scientifically suitable for publication and will be formally accepted for publication once it meets all outstanding technical requirements.

Kind regards,

Roger A. Bannister, PhD

Academic Editor

PLOS ONE

Additional Editor Comments (optional):

Reviewers' comments:

Reviewer's Responses to Questions

**Comments to the Author**

1. If the authors have adequately addressed your comments raised in a previous round of review and you feel that this manuscript is now acceptable for publication, you may indicate that here to bypass the “Comments to the Author” section, enter your conflict of interest statement in the “Confidential to Editor” section, and submit your "Accept" recommendation.

Reviewer #1: All comments have been addressed

Reviewer #2: All comments have been addressed

2. Is the manuscript technically sound, and do the data support the conclusions?

Reviewer #1: Yes

Reviewer #2: Yes

3. Has the statistical analysis been performed appropriately and rigorously? 

Reviewer #1: N/A

Reviewer #2: N/A

4. Have the authors made all data underlying the findings in their manuscript fully available?

Reviewer #1: Yes

Reviewer #2: Yes

5. Is the manuscript presented in an intelligible fashion and written in standard English?

Reviewer #1: Yes

Reviewer #2: Yes

6. Review Comments to the Author

Reviewer #1: (No Response)

Reviewer #2: (No Response)

7. PLOS authors have the option to publish the peer review history of their article (what does this mean?). If published, this will include your full peer review and any attached files.

Reviewer #1: No

Reviewer #2: No

---

## [Editor Report · Acceptance letter]

8 Jul 2021

PONE-D-21-08027R1

Countering reproducibility issues in mathematical models with software engineering techniques: A case study using a one-dimensional mathematical model of the atrioventricular node

Dear Dr. Schölzel:

I'm pleased to inform you that your manuscript has been deemed suitable for publication in PLOS ONE. Congratulations! Your manuscript is now with our production department.

Kind regards,

on behalf of

Dr. Roger A. Bannister

Academic Editor

PLOS ONE